# Modelling of Coupled Hydro-Thermo-Chemical Fluid Flow through Rock Fracture Networks and Its Applications

**Chaoshui Xu [1],\*, Shaoqun Dong [2], Hang Wang [1], Zhihe Wang [1,3], Feng Xiong [4], Qinghui Jiang [5], Lianbo Zeng [6], Leon Faulkner [7], Zhao Feng Tian [8] and Peter Dowd [1]**

[1] School of Civil, Environmental and Mining Engineering, University of Adelaide, Adelaide, SA 5005, Australia; Hang.Wang@adelaide.edu.au (H.W.); wzh9074@hotmail.com (Z.W.); Peter.Dowd@adelaide.edu.au (P.D.)
[2] College of Science, China University of Petroleum, Beijing 102249, China; dshaoqun@163.com
[3] Institute of Deep Earth Sciences and Green Energy, College of Civil and Transportation Engineering, Shenzhen University, Shenzhen 518060, China
[4] Faculty of Engineering, China University of Geosciences (Wuhan), Wuhan 430074, China; whufengx@163.com
[5] School of Civil Engineering, Wuhan University, Wuhan 430072, China; Jqh1972@whu.edu.cn
[6] College of Geoscience, China University of Petroleum, Beijing 102249, China; lbzeng@sina.com
[7] Environmental Copper Recovery Pty Ltd., Kapunda, SA 5373, Australia; lfaulkner@envirocopper.com.au
[8] School of Mechanical Engineering, University of Adelaide, Adelaide, SA 5005, Australia; Zhao.Tian@adelaide.edu.au
\* Correspondence: Chaoshui.Xu@adelaide.edu.au

**Abstract:** Most rock masses contain natural fractures. In many engineering applications, a detailed understanding of the characteristics of fluid flow through a fractured rock mass is critically important for design, performance analysis, and uncertainty/risk assessment. In this context, rock fractures and fracture networks play a decisive role in conducting fluid through the rock mass as the permeability of fractures is in general orders of magnitudes greater than that of intact rock matrices, particularly in hard rock settings. This paper reviews the modelling methods developed over the past four decades for the generation of representative fracture networks in rock masses. It then reviews some of the authors' recent developments in numerical modelling and experimental studies of linear and non-linear fluid flow through fractures and fracture networks, including challenging issues such as fracture wall roughness, aperture variations, flow tortuosity, fracture intersection geometry, fracture connectivity, and inertia effects at high Reynolds numbers. Finally, it provides a brief review of two applications of methods developed by the authors: the Habanero coupled hydro-thermal heat extraction model for fractured reservoirs and the Kapunda in-situ recovery of copper minerals from fractures, which is based on a coupled hydro-chemical model.

**Keywords:** discrete fracture network; modelling of coupled hydro-thermo-chemical fluid flow; geothermal application; in-situ recovery of minerals

## 1. Introduction

In general, for engineering applications, a rock mass can be regarded as the assemblage of two major components: intact rock and fractures/discontinuities. The intact rock component refers to cemented mineral grains plus the voids between grains, while the fractures are the discontinuities between intact rocks where separation boundaries are clearly observable with the naked eye. In this framework, potential micro-cracks existing within the "intact" rock structure are not explicitly identified and are treated only as part of the "intact" matrix structure. For fluid flow through such a rock mass, discontinuities are usually the major fluid conduits through the system as, in general, the permeability of fractures could be orders of magnitude greater than that of the rock matrix (intact rock),

particularly in hard rock settings. Fractures in general make up only a small fraction of the total volume of a rock mass, but they play a decisive role in determining the flow behaviours of the rock mass.

For numerical modelling of fluid flow through a rock mass, the intact rock component is normally represented by a continuous porous medium for which Darcy's law is commonly used as the governing equation to solve the flow problem [1]. For flow through fractures, the solution framework is much more complex and depends on the available information and other performance requirements such as accuracy and computation costs. If the geometries of the fractures and the fracture network within the rock mass are known then, at least in theory, the Navier-Stokes governing equations can be solved for the flow through the fracture network. In this approach, intact rocks and discontinuities are modelled using two separate domains and there will, therefore, be exchange terms between the two domains in the numerical solution framework. This approach is attractive as both components are properly represented in the solution system. However, when fractures and the fracture network are very complex, the solution may rapidly become intractable. There are several difficult issues. Fracture apertures are normally at micron-scales, but fracture extents in engineering applications typically range from the scale of metres to hundreds of metres and the domain size of the model could be up to the scale of kilometres. This contrast in scales makes it impossible to use an element size small enough for accurately meshing the fracture apertures. One simplification is to apply the smooth parallel plate assumption for fractures so that they can be modelled using two-dimensional elements. However, the serious limitations in this assumption are well documented. In general, the walls of rock fractures are not smooth, and walls of the same fracture are not parallel. In addition, the parallel plate model, expressed in the form of the cubic law, is only applicable in low Reynolds number flow regimes, for which the flow is assumed to be laminar and linear [2–4]. Different fracture intersections and their influences on fluid flow further complicate the issues [5]. There are numerous published works in which different correction factors were used for the roughness of fracture walls and for non-linear flow behaviours at high Reynolds numbers [1,6–8]. Even with this two-dimensional simplification, the solution can still become intractable if there is a significant number of fractures in the system, which is often the case in large-scale engineering applications such as groundwater flow modelling, enhanced geothermal systems, or in-situ recovery of minerals. A further simplified approach to address this problem is to use the equivalent continuous porous media (ECPM) approach, in which the flow through fractures is modelled using a substituted continuous porous medium with directional permeability tensors equivalent to those of the corresponding fracture network [9,10]. This includes different variations such as the dual-porosity/dual-continuum (DP/DC) approach, the multiple interacting continua (MINC) model [11], and the stochastic continuum (SC) or fractured continuum (FC) model [12–14]. This approach relies on the assumption that the equivalent permeability of the fracture network within a specified volume can be found, which in practice may be difficult as the representative elementary volume (REV) of a fracture network may not exist [1,9,15]. Another approach to address this problem is to use the equivalent pipe network model in which flow through the fracture network is modelled as flow through a pipe network, which is constructed on the basis of the topology of a fracture network with equivalent permeability [16–18]. Non-linear flow can also be incorporated into the pipe network model by using a friction factor [19]. Although this approach significantly simplifies the solution system and significantly improves the computational efficiency, it also has its limitations; the main one being that the result is sensitive to the constructed pipe network, which in general does not have a unique solution for a complex fracture network.

The modelling of fluid flow through a rock mass, as discussed above, is only possible when the geometries of the fractures and the fracture network within the rock mass are completely known. In reality, however, this is almost an impossible as three-dimensional fractures are not observable or measurable at a scale meaningful for engineering applica-

tions. Powerful 3D scanning using techniques such as X-ray tomography can only handle samples of limited size and much smaller than that required for engineering models due to high resolution required to identify fractures with micron-scale apertures. The construction of a 3D fracture network within a large block of rock through cut slices is possible [20] but this is a costly process that is extremely time-consuming and is difficult to adapt to all applications. The most common approach to constructing a fracture network within a rock mass is via simulations, resulting in discrete fracture network (DFN) models [21]. Although there are deterministic models, most DFN models are stochastic and are generated on the assumption that key fracture properties follow certain probabilistic distributions, which can be inferred from mapping the observable part of the fracture system, either in the form of drill-core surfaces or two-dimensional exposed rock faces. Fracture mapping of drill cores is essentially one-dimensional in which orientations of fractures (dip directions and dip angles) can be measured using oriented cores and fracture spacing can be used to estimate the fracture density in the drill core direction. Fracture mapping of exposed rock faces can provide much more information than that from drill cores. For example, mapping based on a fan of scanlines can be used to map fracture density in different directions (e.g., [22]). Mapping based on window sampling can provide important information related to fracture trace length, which is essential for inferring the statistical distributions of the sizes of 3D fractures [21,23–25]. Note that fracture mapping data based on drill cores or exposed rock faces are always biased as they are only measured in a certain direction or in certain orientations of planes; mapping in all representative directions and 3D planes is, in general, impossible. Some publications have attempted to correct distribution parameters taking into account the biases introduced by fracture mapping in a limited number of directions (e.g., [26,27]). For applications such as enhanced geothermal systems, in which the major fracture network is created by stimulating fractures, seismic event points are a useful source of information about the fracture network that can be used in constructing the 3D fracture model for the rock mass (e.g., [28,29]).

Over the past three decades, many different approaches have been developed for the stochastic generation of DFNs. Each of these has assumptions and a set of required statistical parameters. In general, they can be classified into two different categories, namely stochastic geometry (explicit) and multiple-point statistics (implicit); see [30,31]. The former simulates each fracture on the basis of its geometrical features such as shape, size, and orientation; the latter simulates the entire fracture system based on points defined in space at a given resolution. Most DFN modelling techniques belong to the first category, including the object based (e.g., discs, polygons) approach (e.g., [21,32]), the Poisson plane approach (e.g., [33,34]), and the Boolean modelling approach, with the first being the simplest to implement and the most versatile. For spatially uncorrelated fractures, the fracture density for the object-based approach can be defined by the Poisson model. If the fractures are spatially correlated, non-homogeneous or cluster models can be used [21]. A geostatistical model has also been developed to model the spatial variability of fracture density [35]. A fracture system is commonly modelled as a combination of different sets of fractures, with each set modelled using its own distribution parameters. Fracture sets in a DFN system are usually considered to be independent although correlations can be included using techniques such as hierarchical modelling [14] or plurigaussian modelling [31]. For some applications, in order to produce more realistic fracture models, additional considerations such as termination probabilities are incorporated into the DFN generation process.

This paper provides a review of key DFN modelling techniques developed over the past three decades. It then reviews some recent developments from an international research group comprising the authors of this paper. These developments are in modelling fluid flow through a single fracture and through fracture networks considering challenging issues such as fracture wall roughness, inertial effects at different Reynolds numbers, non-linearity features of fluid flow through fractures, fracture intersections, and fracture networks. Finally, it discusses two applications of coupled hydro-thermal-chemical (HTC) flow modelling for DFN. The first is the extraction of geothermal heat from the Habanero

enhanced geothermal system and the second is the Kapunda project for the in-situ recovery of copper.

## 2. Modelling of Fracture Networks in Rock Masses

Rock fracture modelling can be broadly divided into two areas: deterministic and stochastic (e.g., [36]). Deterministic methods identify fractures based on their physical signatures visible in detections such as X-ray tomography or 3D seismic surveys. The accuracy of these methods depends on the data/detection resolution and they are therefore typically used to model fractures at a specific scale, e.g., large-scale geological fault interpretation using 3D seismic data, or small-scale fracture detection in core samples using X-ray computed tomography [37]. Stochastic methods are commonly used for engineering-scale applications in which a subsurface fracture system is modelled using a stochastic process informed by prior information of the system [38]. Stochastic rock fracture modelling has achieved increasingly wider applications in recent years, particularly in areas such as oil and gas exploitation, extraction of geothermal energy, mining engineering, and hydrogeological engineering.

Dong et al. [36,39,40] systematically reviewed published stochastic fracture modelling methods and classified the methods into five categories: (1) fracture modelling based on spatial sub-division (see Figure 1a–e); (2) discrete fracture network modelling (see Figure 1f–s); (3) fracture modelling based on geostatistics (see Figure 1t); (4) fracture modelling based on multi-point statistics (see Figure 1u); and (5) fracture modelling based on iteration of fractal characteristics (see Figure 1v,w). Another category of recent developments not covered in the reviews mentioned above is a suite of DFN modelling methods that specifically consider the signature of a developing fracture system in terms of micro-seismic events detected during a fracture stimulation process (see Figure 1x).

Fracture modelling based on spatial sub-division can be used to simulate relatively simple fracture systems, such as regular regional tectonic fractures and basalt columnar joints. The four representative models are the unbounded orthogonal model [33,41] shown in Figure 1a, the bounded orthogonal model [33] shown in Figure 1b, the 3D mosaic model extended from the 2D random Voronoi method [42] in Figure 1c, the 3D mosaic model using the 3D random Voronoi method [43,44] shown in Figure 1d, and the Veneziano model shown in Figure 1e. Even though the Veneziano model can also simulate complex fracture systems, it is not widely used due to its complexity.

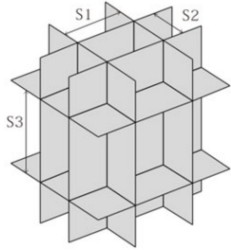

(**a**) Unbounded orthogonal

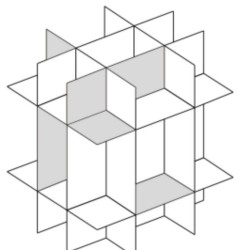

(**b**) Bounded orthogonal

**Figure 1.** *Cont.*

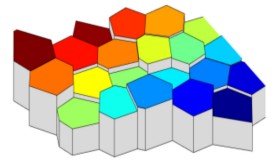

(**c**) 2D to 3D mosaic model

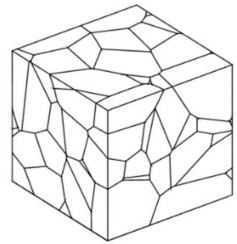

(**d**) 3D mosaic model

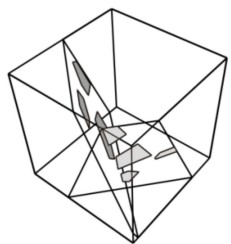

(**e**) Veneziano model

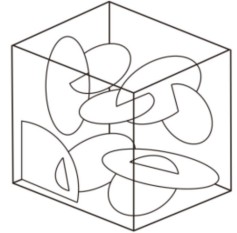

(**f**) DFN model-based MPP

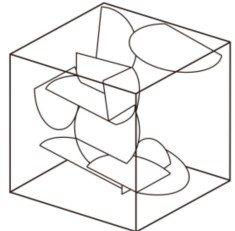

(**g**) Enhanced Baecher model

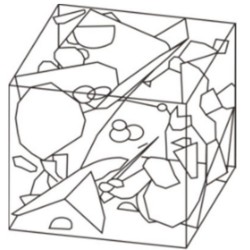

(**h**) BART model

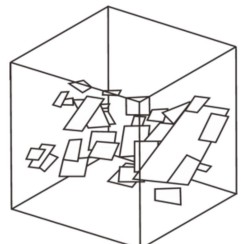

(**i**) Poisson rectangle model

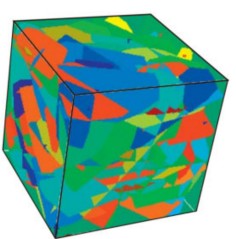

(**j**) Random polygon model

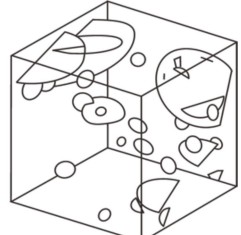

(**k**) Nearest neighbour model

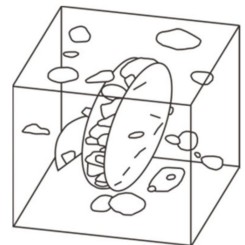

(**l**) War zone model

**Figure 1.** *Cont.*

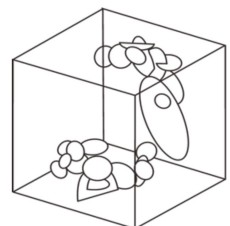

(**m**) Levy–Lee fractal model

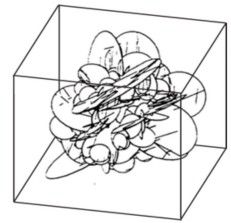

(**n**) Parent–daughter model

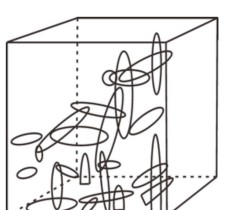

(**o**) BFFN model

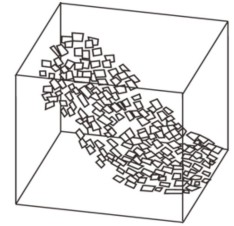

(**p**) Non-planar zone model

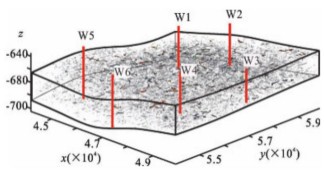

(**q**) Density-controlled DFN model

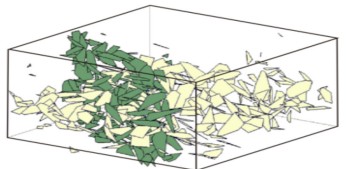

(**r**) GEOFRAC model

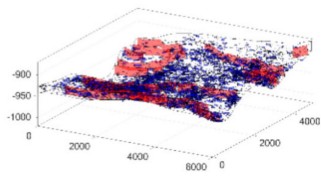

(**s**) Multi-scaled fracture network model

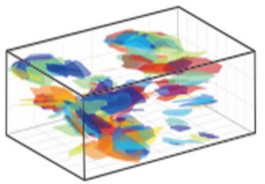

(**t**) Modelling based on Geostatistics

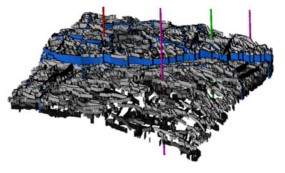

(**u**) Multi-point statistics model

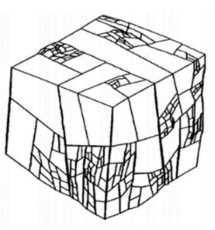

(**v**) 3D IFS model

**Figure 1.** *Cont.*

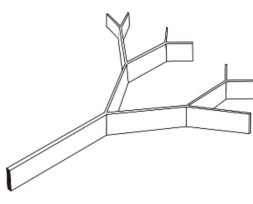

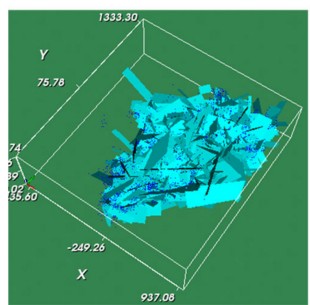

(**w**) 3D L-system model

(**x**) DFN model conditioned to a seismic event point cloud based on algorithms such as MCMC, clustering, RANSAC, and PANSAC

**Figure 1.** Schematic diagrams of stochastic fracture modelling methods [21,29,32–34,36,39–41,43,45–49]. (**a**–**e**) fracture modelling based on spatial subdivision, (**f**–**s**) discrete fracture network modelling, (**t**) fracture modelling based on geostatistics, (**u**) fracture modelling based on multi-point statistics, (**v**,**w**) fracture modelling based on iteration of fractal characteristics, (**x**) fracture modelling conditioned to a seismic event point cloud.

General discrete fracture network (DFN) modelling based on marked point processes was independently proposed by Baecher et al. [50] and Barton [51], with a model example shown in Figure 1f. Its wider application to rock engineering was promoted in the 1980s by several research groups (e.g., [41,52,53]). This modelling method employs a marked point process (MPP) to simulate the fracture network using circular or elliptical disks to represent fractures [21,34,41,54]. The point process determines the centre locations of fractures while the associated marks determine fracture properties such as fracture size, orientation, and aperture [21,31,46]. A Poisson point process (homogeneous point process) is the most commonly used, although non-homogeneous point processes can also be used to model more complex systems and to take into account spatial correlations of fractures [21]. Different correlation structures can also be considered in DFN modelling [55,56]. The fracture system is normally modelled using several fracture sets. The workflow to generate one set of fractures (containing only four fractures for illustration purposes) using the MPP modelling method is shown in Figure 2. The process is repeated for other fracture sets and the union of all generated fractures leads to a complete DFN model for the entire fracture system. Different sets of fractures are normally modelled independently, although correlations between different sets of fractures can also be incorporated using modified approaches [21,31,57].

Since its introduction, many modified DFN models have been developed to make DFN more suitable for simulating different subsurface fracture systems. For example, many models have been developed with modified marked processes, especially for modelling different fracture shapes: the enhanced Baecher model [58] shown in Figure 1g (fractures can clip each other); the Baecher algorithm revised terminations (BART) model [33,41] shown in Figure 1h (fractures may have random sizes and shapes); the Poisson rectangle model [33] shown in Figure 1i (fractures are set as rectangles); and the random polygon model [21,46] shown in Figure 1j (fractures are set as random polygons). Models have also been developed with different point processes: the nearest neighbour model [33,34] shown in Figure 1k (this uses non-stationary Poisson point processes, see [21]); the war zone model [34] shown in Figure 1l; the Levy-Lee fractal model [33,41] shown in Figure 1m; the parent-daughter model [21,59] shown in Figure 1n; the binary fractal fracture network (BFFN) model [60,61] shown in Figure 1o; the non-planar zone model shown in Figure 1p; and the density-controlled DFN model shown in Figure 1q (which uses both a density-controlled Poisson point process and the random polygon models). In addition, some models have replaced the MPP by other methods: the GEOFRAC model shown in Figure 1r, which is based on Poisson line (2D) or Poisson plane (3D) processes (an improvement on

the Veneziano model). In practice, DFN can also be combined with deterministic modelling methods to simulate multi-scale fracture networks, see the example shown in Figure 1s.

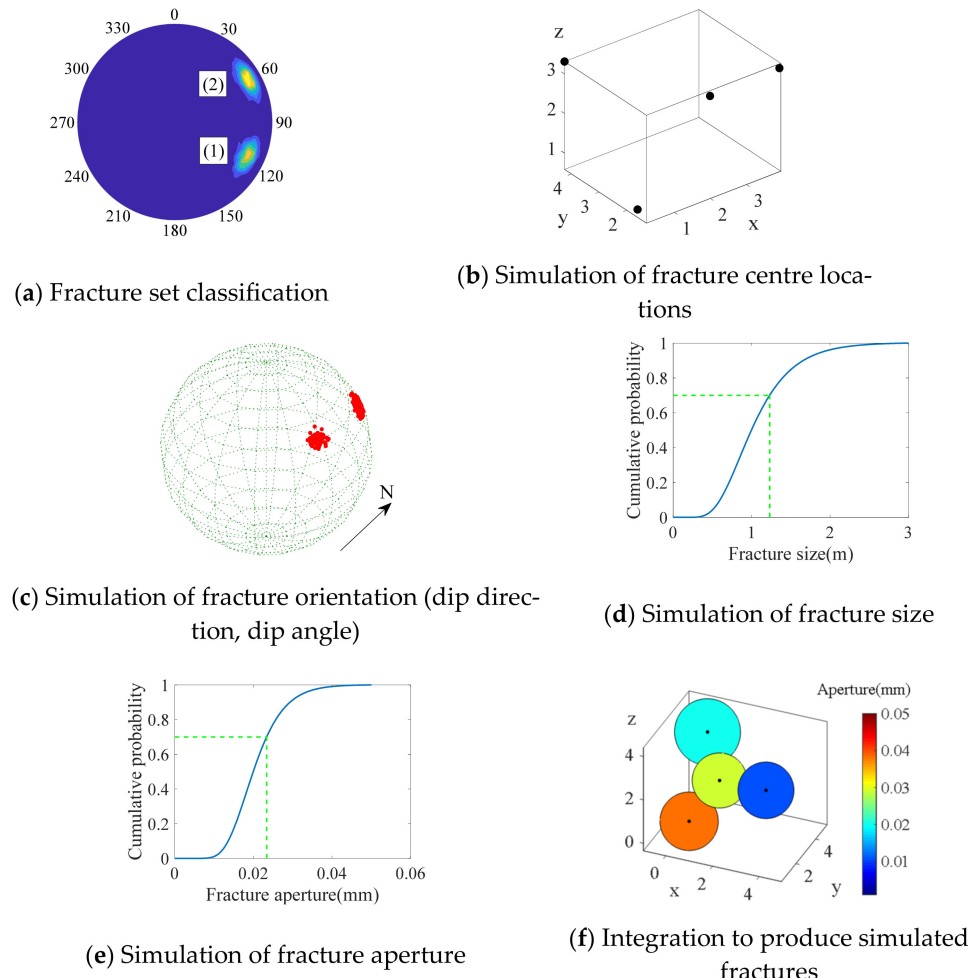

(**a**) Fracture set classification

(**b**) Simulation of fracture centre locations

(**c**) Simulation of fracture orientation (dip direction, dip angle)

(**d**) Simulation of fracture size

(**e**) Simulation of fracture aperture

(**f**) Integration to produce simulated fractures

**Figure 2.** Workflow of the MPP modelling method. (**a**) Fracture set classification, (**b**) Simulation of fracture centre locations by a point process, (**c**–**e**) Simulations of fracture properties such as strikes (or dip directions), dips, sizes, and apertures, (**f**) Integration of fracture locations and properties to produce final simulated fractures.

Fracture modelling based on geostatistics is often done within the framework of the DFN model based on the MPP approach. Variograms are used to model fracture density and fracture positions are simulated based on local density variations to reflect the spatial correlation of fractures [49]. In addition, Dowd et al. [31] generated fracture networks by combining marked point processes with simulated fracture locations using truncated pluri-Gaussian simulation to include spatial correlations between different fracture sets. Geostatistical modelling can be used not only to replace the point process for the simulation of fracture locations, but also to replace the mark process for the simulation of fracture properties. Koike et al. [49,62] used the indicator transformation and geostatistics to simulate the strikes and dips of fractures for their GEOFRAC model (Figure 1t); note that this model is different to the GEOFRAC model developed in Ivanova et al. [43] (see Figure 1r).

Fracture modelling based on multi-point statistics (MPS) can characterize complex spatial structures. Dowd et al. [31] used the SNESIM algorithm to simulate a 2D fracture profile using mapped fractures from the Yucca Mountain nuclear repository and showed that multi-point statistical simulation can simulate complex fracture networks. Moham-madmoradi [63] and Jia et al. [64] improved the FILTERSIM algorithm to simulate induced

fracture networks. Liu et al. [65] used the SIMPAT algorithm to model two-dimensional fracture networks resulting in good reproductions of the fracture system. A common concern with MPS approaches is the stationarity issue. To overcome this problem, Chugunova et al. [66] proposed a non-stationary MPS simulation method using remotely sensed data and the resulting model is shown in Figure 1u. In addition to the stationarity problem, MPS commonly suffers from the lack of sufficient training data to model the fracture system.

Fracture modelling by iterating fractal characteristics takes advantage of the fractal nature of some fracture networks, such as fracture networks induced by hydraulic fracturing. Commonly used methods include the iterated function system (IFS) and the L-System, which can be used to generate self-similar fractals. Acuna and Yortsos [47] used IFS to model 3D fracture network systems as shown in Figure 1v. The L-System generates geometric objects similar to a tree object in which branches extend from a trunk. Wang et al. [48] developed a discrete fractal-fracture network modelling based on L-System to simulate complex fracture networks in shale reservoirs (see Figure 1w) and verified the approach using numerical simulations. This type of method is only suitable for modelling fracture networks with obvious fractal characteristics.

Recently, a suite of DFN modelling methods was developed to condition the discrete fracture network model by 3D seismic event point clouds; see an example shown in Figure 1x. The context of these developments is for applications such as enhanced geothermal reservoir characterisation, stimulated reservoir development for unconventional oil and gas exploitation, and the in-situ recovery of minerals. In these applications, during the stimulation process of the reservoir, fracture development produces seismic events that can be detected using an array of geophones. These events contain important information about the fracture system and these newly developed methods take advantage of this signature information to develop a DFN system conditioned to the events and hence generate a more realistic representation of the fracture system. The methods include a Markov Chain Monte Carlo (MCMC) approach [28], a clustering approach [67,68], the RANSAC approach [69], and the PANSAC approach [29]. These are discussed further in Section 4.1.

Fracture modelling based on spatial subdivision is simple but, in general, it is not sufficient to represent the complexity of the structure of a fracture network. Fracture modelling based on geostatistics is often not truly independent and it more or less depends on DFN. For example, geostatistics can be used for modelling fracture density but, once that is done, a DFN approach (MPP) is required to generate the fractures. In practice, fracture modelling based on multi-point statistics is difficult because of the usual lack of suitable training data. Fracture modelling based on iterating fractal characteristics is only suitable for specific fracture networks and the scope of applications is relatively limited. In comparison, DFN modelling using MPP is the most widely used approach largely because it is simple to implement, versatile, and can model very complex fracture systems. In addition, it can also easily integrate fracture mapping data in a more logical manner [25].

## 3. Modelling Fluid Flow through DFNs

This section summarises some recent developments from an international research group comprising the authors of this paper. These developments are in modelling fluid flow through a single fracture and through fracture networks. The work covers a wide spectrum of topics in this area, including challenging issues such as fracture wall roughness, inertial effects at different Reynolds numbers, non-linearity features of fluid flow through fractures, fracture intersections, and fracture networks. A general numerical modelling framework for DFN using COMSOL and FracSim3D is discussed first.

### 3.1. Using COMSOL and FracSim3D to Model Fluid Flow through DFNs

COMSOL is a general-purpose finite element numerical simulation software package for solving multi-physics problems. A simple approach to modelling fluid flow through fracture networks in COMSOL is to use Darcy's Law, for which fracture properties, includ-

ing aperture, porosity, and permeability, need to be defined. The aperture and porosity of fractures can be defined as constants or can be specified in a parametric form (e.g., spatially variable). For fracture permeability, either a user-defined value or a Cubic Law model with user-defined aperture and roughness factor can be used. Different properties can also be assigned to each fracture in COMSOL.

Complex fracture geometries can be imported into COMSOL and the flow system can be analysed using a full 3D model. However, such a model could quickly become intractable for large, complex fracture networks. A simple approach to addressing this issue, as discussed above, is to represent fractures by 2D planar surfaces. The planar geometry of each fracture must be defined in order to generate the mesh required for the numerical simulation. For a fractured rock mass, the definition of fracture geometries is in the form of a representative DFN model generated from the available information about the fracture network. FracSim3D [21] is a versatile and comprehensive DFN modelling tool based on marked point processes which can be used to generate complex fracture networks. FracSim3D is freeware available on request from the first author. The DFN model generated in FracSim3D contains geometrical data, including fracture orientation and the vertex coordinates of each fracture polygon, which define the fracture geometries required in COMSOL. Figure 3 shows a DFN model generated by FracSim3D, which is re-produced (using the geometrical data) and meshed in COMSOL. In this example, the DFN is cropped by a 35 × 35 × 35 m cubic block.

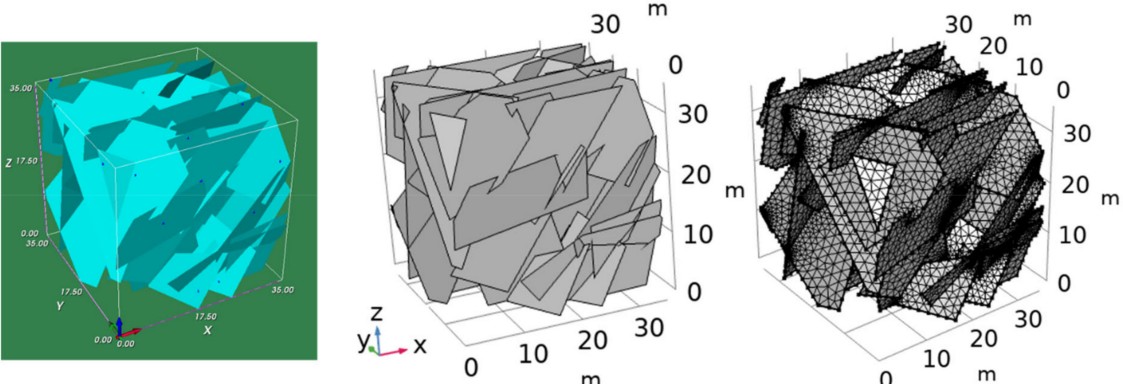

**Figure 3.** A DFN model generated in FracSim3D and re-produced and meshed in COMSOL. Fracture centres are generated within a 35 × 35 × 35 m cubic block and fractures are cropped by this block.

As discussed above, the conductivity of fractures is normally orders of magnitude higher than that of the rock matrices, particularly in hard rock settings. A simple approach to fluid flow modelling in this context is to focus on flow through the fracture network, although the flow through the rock matrices can also be incorporated into this solution framework using the methods discussed in the previous section. Although fractures are distributed in 3D space, fluid flow through a DFN is essentially a 2D problem since fractures are represented by 2D surfaces together with their respective conductivity values. As an example of solving such a system, constant hydraulic pressure boundaries are assumed, which apply to all edges of fractures intersecting the six boundary faces of the cubic block (Figure 3). Assuming that a constant pressure gradient is applied in the x direction (Equation (1) and Figure 3), the hydraulic pressure at any point on the block boundary faces can be calculated by Equation (2), which is then applied to the edges of boundary-intersecting fractures as the constant pressure boundary conditions:

$$\nabla P = \frac{P_2 - P_1}{x_2 - x_1} \tag{1}$$

$$P = P_1 + \frac{(P_2 - P_1)}{x_2 - x_1}(x - x_1) \tag{2}$$

where $P_1$ and $P_2$ are the pressures at the faces $x = 0$ m and $x = 35$ m of the block. Using $P_1 = 100$ Pa, $P_2 = 0$ Pa and the Cubic Law for fracture permeability with a uniform aperture of 0.5 mm, porosity of 0.9, and roughness factor of 1, the simulated flow model for the example DFN is shown in Figure 4.

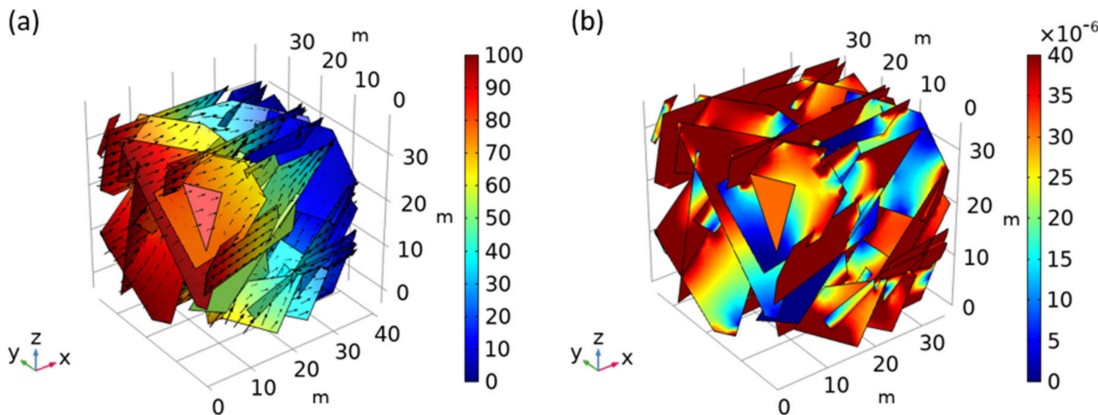

**Figure 4.** (**a**) Simulated pressure (colour-coded in Pa) and fluid flow velocity field (arrow); (**b**) Magnitude of flow velocity (colour-coded in m/s).

The total flow through each block face can be calculated by integrating the flow over the fracture edges at the block boundary. The results are summarized in Table 1. In this example, the flows through the left, behind, and top faces are inflows, while the flows through the other three faces are outflows. The total inflow is $5.415 \times 10^{-6}$ m³/s and the total outflow is $5.423 \times 10^{-6}$ m³/s, with a calculation error of 0.14%.

**Table 1.** Flow through DFN block boundaries with pressure gradient in the $x$ direction.

| Pressure Gradient Direction, Flow | Block Boundaries | | | | | |
|---|---|---|---|---|---|---|
| | **Left** | **Right** | **Front** | **Behind** | **Bottom** | **Top** |
| | (x-) | (x+) | (y-) | (y+) | (z-) | (z+) |
| x, Q (m³/s) | $2.90 \times 10^{-6}$ | $4.75 \times 10^{-6}$ | $-1.38 \times 10^{-9}$ | $-1.98 \times 10^{-7}$ | $2.32 \times 10^{-6}$ | $6.75 \times 10^{-7}$ |

A simplified flow modelling approach for fractured rock masses is via equivalent continuous porous media (ECPM). For example, if the DFN example (Figure 3) is to be treated as an ECPM, the permeability tensor of this DFN model can be derived by simulating fluid flow through the DFN model with the pressure gradient applied in the x, y and z directions respectively. Equation (3) can be obtained from Darcy's Law, where $Q_x$, $Q_y$, and $Q_z$ denote the directional fluid flux (m³/s) through the DFN model when pressure gradients of $(\frac{\partial p}{\partial x}, 0, 0)$, $(0, \frac{\partial p}{\partial y}, 0)$, and $(0, 0, \frac{\partial p}{\partial z})$ are applied, respectively; $A$ is the cross-sectional area of the block, $k$ with footnotes represents the corresponding permeability tensor component and $\mu$ is the fluid viscosity. Table 2 lists the simulation results for this example with the same pressure gradient applied in the x, y, and z directions, respectively.

$$\left[ \frac{Q_x}{A} \ \frac{Q_y}{A} \ \frac{Q_z}{A} \right] = \begin{pmatrix} q_{xx} & q_{xy} & q_{xz} \\ q_{yx} & q_{yy} & q_{yz} \\ q_{zx} & q_{zy} & q_{zz} \end{pmatrix} = -\frac{1}{\mu} \begin{pmatrix} k_{xx} & k_{xy} & k_{xz} \\ k_{yx} & k_{yy} & k_{yz} \\ k_{zx} & k_{zy} & k_{zz} \end{pmatrix} \begin{pmatrix} \frac{\partial p}{\partial x} & 0 & 0 \\ 0 & \frac{\partial p}{\partial y} & 0 \\ 0 & 0 & \frac{\partial p}{\partial z} \end{pmatrix} \tag{3}$$

**Table 2.** Flow through DFN block boundaries with pressure gradient in x, y, and z directions.

| Pressure Gradient Direction, Flow | Block Boundaries | | | | | |
|---|---|---|---|---|---|---|
| | Left | Right | Front | Behind | Bottom | Top |
| | (x-) | (x+) | (y-) | (y+) | (z-) | (z+) |
| x, Q (m$^3$/s) | $2.90 \times 10^{-6}$ | $4.75 \times 10^{-6}$ | $-1.38 \times 10^{-9}$ | $-1.98 \times 10^{-7}$ | $2.32 \times 10^{-6}$ | $6.75 \times 10^{-7}$ |
| y, Q (m$^3$/s) | $-1.15 \times 10^{-8}$ | $-5.72 \times 10^{-7}$ | $2.40 \times 10^{-6}$ | $3.56 \times 10^{-6}$ | $6.23 \times 10^{-7}$ | $1.70 \times 10^{-8}$ |
| z, Q (m$^3$/s) | $1.03 \times 10^{-6}$ | $1.85 \times 10^{-6}$ | $-6.24 \times 10^{-7}$ | $9.80 \times 10^{-7}$ | $7.01 \times 10^{-6}$ | $4.62 \times 10^{-6}$ |

The directional fluid flow $Q_x$, $Q_y$, and $Q_z$ can be seen in Table 2. For example, the x component of $Q_x$ (with applied pressure gradient in the x direction) is the flow rate through the left (x-) and right (x+) faces shown in the first two columns of the first row. Note that as a DFN is, in general, intrinsically anisotropic and inhomogeneous, the flows through the two parallel faces of the same block are usually different, and the permeability tensor is also not symmetric, i.e., $k_{ij} \neq k_{ji}$. Therefore, average values are used to derive the equivalent permeability tensor for the DFN model. Using the results in Table 2 and Equation (3), the permeability tensor for this DFN example can be calculated and the results are shown in Equation (4). The principal permeability tensor for this fracture network can also be evaluated as summarized in Table 3.

$$k = \begin{pmatrix} 1.102 & -0.057 & 4.242 \\ -0.565 & 8.583 & 0.718 \\ 4.242 & 0.718 & 16.764 \end{pmatrix} \times 10^{-12} \qquad (4)$$

**Table 3.** Principal permeability tensor derived for the DFN model.

| Principal Permeability | Permeability (m$^2$) | Principal Direction (°) | |
|---|---|---|---|
| | | Trend | Plunge |
| $k_{11}$ | $1.903 \times 10^{-12}$ | 4.4 | 62.1 |
| $k_{22}$ | $9.510 \times 10^{-13}$ | 135.4 | 19.1 |
| $k_{33}$ | $7.823 \times 10^{-13}$ | 232.4 | 19.4 |

Note: Principal permeability directions are based on a coordinate system of positive x being northing and positive z being downward vertical.

Using COMSOL and FracSim3D to model fluid flow through a DFN provides a simple approach to studying the hydraulic property of a fractured rock mass when statistical information about the fractures is available. In the approach demonstrated, fractures are modelled as 2D surfaces and details of the fracture properties, such as spatially variable fracture apertures and fracture surface roughness, are not modelled explicitly (see next section for explicit modelling of these properties). This approach reduces model complexity and significantly improves computational efficiency, which enables it to be used to deal with large number of fractures in a DFN model in engineering applications. However, as the number of fractures increases, the number of fracture intersections increases significantly, which may increase the computational cost significantly due to the complexity of modelling flow around fracture intersections (see Section 3.3).

The above example uses a small block to illustrate the process of estimating the permeability tensor of a DFN model using COMSOL. Such an estimation can also be performed using several other software packages, such as FRACMAN [70] and DFNWorks [71]. To use the ECPM approach at the reservoir scale, the region of interest can be sub-divided into blocks with their estimated equivalent permeability tensors. This approach is based on the assumption that sub-blocks satisfy the representative elementary volume requirement as described in Section 1. This could sometimes be challenging due to the inherent heterogeneity of fracture networks and therefore a certain degree of approximation will have to be used in practice.

### 3.2. Fluid Flow Modelling in a Single Fracture Considering Roughness, Void Structure, and Inertia Effects

Successful modelling of fluid flow in fracture networks requires a comprehensive understanding of fundamental theories of flow in a single fracture and the potential errors associated with the assumptions and simplifications made (cf. Section 3.1) to assess the real flow process [9,72–75]. Several recent studies have attempted to incorporate fracture-scale variability into the fracture network models to test the effect of aperture heterogeneity on the overall flow [76,77]. As computing power continues to grow, incorporating the aperture heterogeneity of each discrete fracture into more complex DFN models may become a common approach in future studies for more accurate quantification of flow behaviours in fractured rock formations. Therefore, it is important to understand the flow behaviours of a single fracture, and to examine the performance of different conceptual flow models on describing different flow scenarios.

The most common case of modelling flow in fractures is for incompressible single-phase flow at steady state, with fracture walls often being considered to have no-slip and no-flow features [9,78–83]. The governing equations for this scenario are the three-dimensional (3-D) Navier-Stokes equations (NSE) with mass conservation [6,8,84], as given by:

$$\rho(\boldsymbol{u}\cdot\nabla)\boldsymbol{u} - \mu\nabla^2\boldsymbol{u} + \nabla P = 0 \tag{5a}$$

$$\nabla\cdot\boldsymbol{u} = 0 \tag{5b}$$

where $\rho$ is the fluid density, $\boldsymbol{u}$ is the velocity, $\mu$ is the dynamic viscosity, and $P$ is the reduced pressure. With the increase in computing power since the early 2000s, numerically solving the NSE in a single rock fracture (at the scale of ~100 mm × 100 mm) has been done extensively [2,6,85–88]. However, accurate numerical analyses of flow problems in fracture networks are still a major challenge due to difficulties in obtaining detailed in-situ fracture void geometry in practice and extremely high computational cost in solving the 3-D NSE in complex fracture networks. A considerable amount of work has been done to develop alternative models, both conceptually and empirically [74,88–90]. Early studies [91–93] considered rock fractures to have smooth and planar walls and, accordingly, the cubic law (CL) was derived from the NSE as:

$$Q = -\frac{Wb^3}{12\mu}\frac{\Delta P}{L} \tag{6}$$

where $Q$ is the volumetric flowrate, $W$ is the fracture width, $b$ is the fracture aperture, and $L$ is the fracture length. Since its introduction, the CL has found broad application in estimating the permeability of a single fracture and an entire fracture network. However, later studies demonstrated that, due to surface roughness, the CL can overestimate the fracture flow by 30–70% [88,94,95], compared to results from both flow experiments and simulations. Modifications are often incorporated to improve flow prediction using the CL. In a recent study, we presented a modified cubic law (MCL) that considers the effects of flow tortuosity, aperture variation, and local roughness on flow behaviours [6]. The MCL enables an improved flow prediction compared with the standard CL, and is given by:

$$Q = \frac{<b_{modified}>^3}{12\mu Flength}W\frac{\Delta P}{L}\left[1 + \left(\frac{\sigma_{b_{modified}}}{<b_{modified}>}\right)^2\right]^{-1.5} \tag{7}$$

where $F_{\text{length}}$ is the length of flow path defined in [6] and $b_{\text{modified}}$ is defined as

$$b_{modified}(x) = T_v^{1/3}\cdot F_t\cdot F_{lr} \tag{8a}$$

$$T_v = \frac{2b(i)^3 b(i+1)^3}{b(i)^3 + b(i+1)^3} \tag{8b}$$

$$F_t = \frac{2\cos\alpha_u\cos\alpha_l}{\cos\frac{\alpha_u-\alpha_l}{2}\left(\cos\alpha_u+\cos\alpha_l\right)} \tag{8c}$$

$$F_{lr} = \left(1 - \frac{\sigma_u+\sigma_l}{2B_s}\right) \tag{8d}$$

where $b(i)$ and $b(i+1)$ are adjacent apertures; $\alpha_u$ and $\alpha_l$ are the inclination angles of upper and lower walls; $\sigma_u$ and $\sigma_l$ are the wall roughness variations of the upper and lower walls; $B_s$ is the segment half aperture. In the proposed MCL, $T_v$ accounts for the effect of aperture variation; $F_t$ is the modification factor for segment tortuosity; and $F_{lr}$ is the modification factor for local roughness (more details can be found in [6]). The performance of the MCL was assessed against numerically solving the NSE in 45 synthetic fractures with different surface roughness and void structures. The predicted flow showed close agreement with the results from the NSE with a mean absolute error of 4.7%, as compared to 26.5% for the standard CL and 9.5% for another published version of the modified cubic law that adopted the "perpendicular aperture" [88]. Despite its effectiveness in improving the standard CL, the MCL fails to consider the in-plane tortuosity, which is an inherent defect for all versions of CL. This hinders its application in fractures with strong flow channelling.

When flow channelling is an important issue, the Reynolds equation (RE, also referred to as the local cubic law) is often used for flow estimation, and takes the following form [84,96,97]:

$$\nabla\cdot\left[\frac{b^3}{12\mu}\nabla P\right] = 0 \tag{9}$$

The RE can be derived from the NSE when considering a slowly varying aperture field and neglecting flow inertia (another approach is to assume that the CL is valid locally throughout the fracture void). Although the RE can capture flow channelling behaviour, the validity of its original version in predicting fracture flow has been questioned by many studies [2,80,97–99]. Due to inaccurate quantification of the effect of aperture variability and not considering the spatial undulation of the fracture mid-surface, the RE is found to overestimate flow by up to 47% when compared with experimental results [88,94]. Many studies have attempted to include pore-scale modifications to improve the standard RE by considering mid-surface variation and applying various averaging schemes for local transmissivity. In a departure from previous approaches, we proposed a non-linear version of the Reynolds equation (NRE) based on a two-dimensional (2-D) perturbation solution (PS) that can accurately estimate the transmissivity at local wedge-shaped cells [7]. The derived PS is an approximate analytical solution to the 2-D NSE, and can be seen as a further extension to the widely adopted CL assumption used locally in the RE [2]. A description of the approach is briefly presented below.

For the void within a fracture, its geometry is widely approximated by simply connecting the measured adjacent elevation points of the top and bottom surfaces, as shown in Figure 5.

A stream function $\psi$ is introduced, which is defined by:

$$u = \frac{\partial\psi}{\partial z}, \; w = -\frac{\partial\psi}{\partial x} \tag{10}$$

The auxiliary condition is expressed by:

$$\Delta p = \int_{x_1}^{x_2} \frac{1}{b}\left(\int_{b_b}^{b_t}\frac{\partial p}{\partial x}dz\right)dx \tag{11}$$

The stream function and pressure difference can be made dimensionless and expressed as an expanded series with a small parameter $\epsilon$:

$$\Psi = \Psi_0 + \epsilon\Psi_1 + \epsilon^2\Psi_2 + O\left(\epsilon^3\right) \tag{12}$$

$$\Delta P = \Delta P_0 + \epsilon \Delta P_1 + \epsilon^2 \Delta P_2 + O\left(\epsilon^3\right) \tag{13}$$

where $\epsilon$ in this study is defined as $\epsilon = \omega/l$; $\omega$ is the dimensionless absolute aperture variation defined as $\omega = |a|/h_m$; $\Psi$ is the dimensionless stream function defined as $\Psi = \psi/q$; $\Delta P$ is the dimensionless pressure difference given as $\Delta P = \Delta p/\Delta p_m$ and $\Delta p_m$ is the pressure difference of flow through a fracture with a uniform aperture defined from the CL as:

$$\Delta p_m = \frac{12\mu l Q}{b_m{}^3} \tag{14}$$

The stream function can be derived by substituting Equation (14) into the dimensionless 2-D NSE. The final PS can be found by inserting the derived stream function into the auxiliary condition. More details can be found in our previous work [7]. The equivalent hydraulic aperture $b_T$ of the wedge can be defined by multiplying the mean aperture $b_m$ by an extra modification factor:

$$b_T = b_m \cdot F_p \tag{15}$$

where $F_p$ is a modification factor based on the perturbation solution (PS) given by $F_p{}^3 = \Delta p_m/\Delta p$. Finally, the proposed NRE is given by:

$$\frac{\partial}{\partial x}\left[b_T{}^3(x,y)\frac{\partial p}{\partial x}\right] + \frac{\partial}{\partial y}\left[b_T{}^3(x,y)\frac{\partial p}{\partial y}\right] = 0 \tag{16}$$

Equation (16) is in a similar form as the Reynolds equation, except that the original measured apparent aperture $b$ is replaced by $b_T$ to incorporate the varying local fracture geometry and inertial effects. Using a finite difference approach similar to [100], the NRE was solved iteratively using MATLAB. The performance of the proposed NRE was tested against flow experiments and flow simulations by solving numerically the three-dimensional NSE for three cases of rock fractures with different void geometries (see Figure 6). The pressure difference obtained from the NRE demonstrates the similar non-linear behaviour as that obtained from the simulations with the increase of the Reynolds number $R$ (see Figure 7). Overall, the mean discrepancy between the proposed model and flow simulations is 5.7% for $R$ ranging from 0.1 to 20, indicating that the proposed NRE captures well the flow non-linearity in rock fractures.

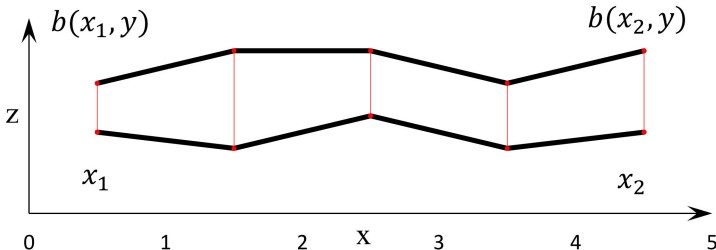

**Figure 5.** Illustration of a series of two-dimensional connected local wedges along the longitude direction $x$; a similar set of connected wedges can also be formed along the latitude direction $y$ (after [7]).

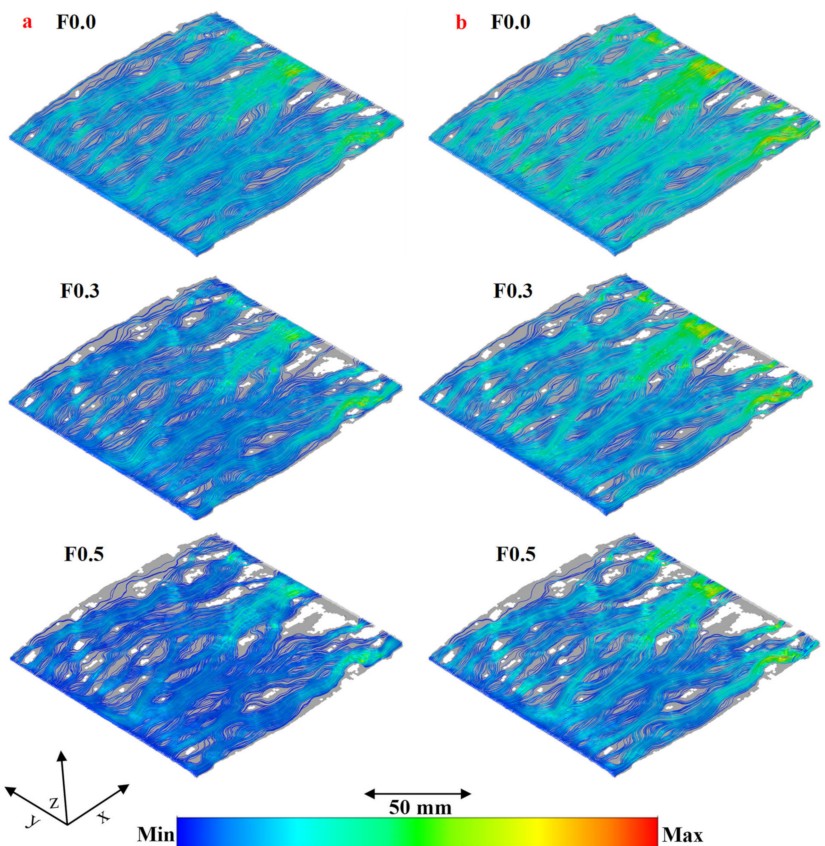

**Figure 6.** Flow streamlines of the three fracture cases (F0.0, F0.3 and F0.5 with different void geometries) for (**a**): *R* = 1 and (**b**): *R* = 20, where blue represents the minimum velocity and red is the maximum velocity. The grey areas are fracture voids with little flow and the white areas are surface contact areas. The inflow *q* for all simulations is along the *x* direction (after [7]).

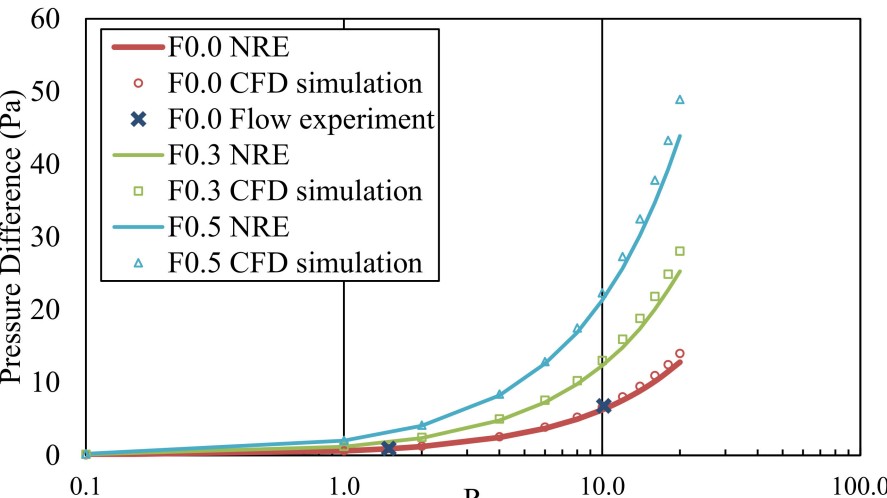

**Figure 7.** Pressure difference $\Delta p$ between the inflow and outflow boundaries of tested fractures for *R* ranging from 0.1 to 20, obtained from flow simulations, experiments, and the proposed NRE (after [7]).

### 3.3. Non-Linear Fluid Flow Modelling in Fracture Intersections

In addition to fluid modelling through single fractures, as discussed above, modelling of flow around fracture intersections is another important component for understanding flow through fracture networks. There are four basic types of fracture intersections that

affect the effective permeability of DFNs, namely, straight fracture intersection (SFI), buck-
ling fracture intersection (BFI), crossing fracture intersection (CFI), and furcating fracture
intersection (FFI), as shown in Figure 8. When fluid passes a fracture intersection, the flow
pressure and flowrate distribution are affected by the fracture intersection angle. The fluid
flow characteristics derived for a single fracture are not directly applicable to the case of
fracture intersections, especially when the non-linear flow regime is considered.

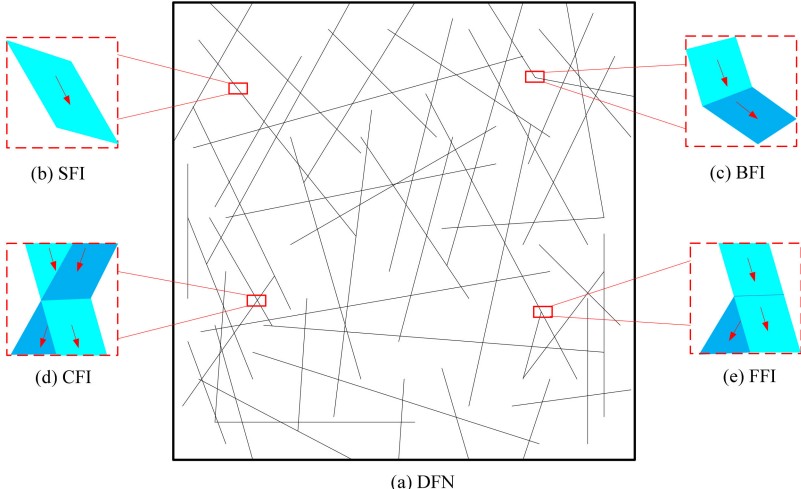

(b) SFI     (c) BFI

(d) CFI     (e) FFI

(a) DFN

**Figure 8.** (**a**) A schematic diagram of a discrete fracture network containing different types of fracture
intersections: (**b**) Straight fracture intersection (SFI), (**c**) Buckling fracture intersection (BFI), (**d**)
Crossing fracture intersection (CFI), and (**e**) Furcating fracture intersection (FFI).

To understand the flow behaviour at fracture intersections, different numerical models
were created based on the geometry of fracture intersections. The Naiver-Stokes equation
was solved for the flow system. A constant velocity condition was used for the inlet
boundary of the fracture intersection, and a constant pressure boundary condition was
used for the outlet. Non-slip boundary conditions were assigned to the fracture walls. In
numerical simulations, the momentum and pressure were solved using a second-order
upwind scheme and a second-order scheme, respectively. The pressure-velocity coupling
was calculated using the SIMPLE algorithm [101]. The solution gradients were evaluated
by direct interpolation using a least squares method at the center of each cell. Steady state
solutions were considered to have converged when the normalized velocity and continuity
residual ratios were less than $1.0 \times 10^{-4}$.

Some examples of the numerical results are illustrated below. Figure 9 shows the
pressure and flow profiles for the case of BFI at hydraulic gradient $J = 0.001$ and $J = 0.3$. For
buckling angles $\alpha$ of 90° and 120°, as the hydraulic gradient increases, eddies appear at
the buckling position of the top surface and the lower part of the bottom surface, causing
the effective flow channel to be narrowed. However, for a buckling angle of 150°, no eddy
can be observed in the fracture, indicating a very low degree of non-linearity. Figure 10
shows the pressure and flow profiles for the case of CFI at hydraulic gradient $J = 0.01$ and
$J = 0.3$. For a crossing angle $\theta$ of 5°, the flow streamlines follow closely the curvatures of
the fracture walls. For crossing angles $\theta$ of 30° and 60°, two eddies emerge in the lower
parts of the fracture, which cause a reduction in the effective area available for flow when
the hydraulic gradient $J$ is 0.3. Figure 11 shows the pressure and flow distribution profiles
for the case of FFI at $J = 0.01$ and $J = 0.3$. Two eddies appear in the lower parts of the
fracture with the furcating angle $\gamma$ of 20° and 75°. For furcating angles of 40°, no eddy can
be observed within the fracture, even when the flow is in the Forchheimer regime.

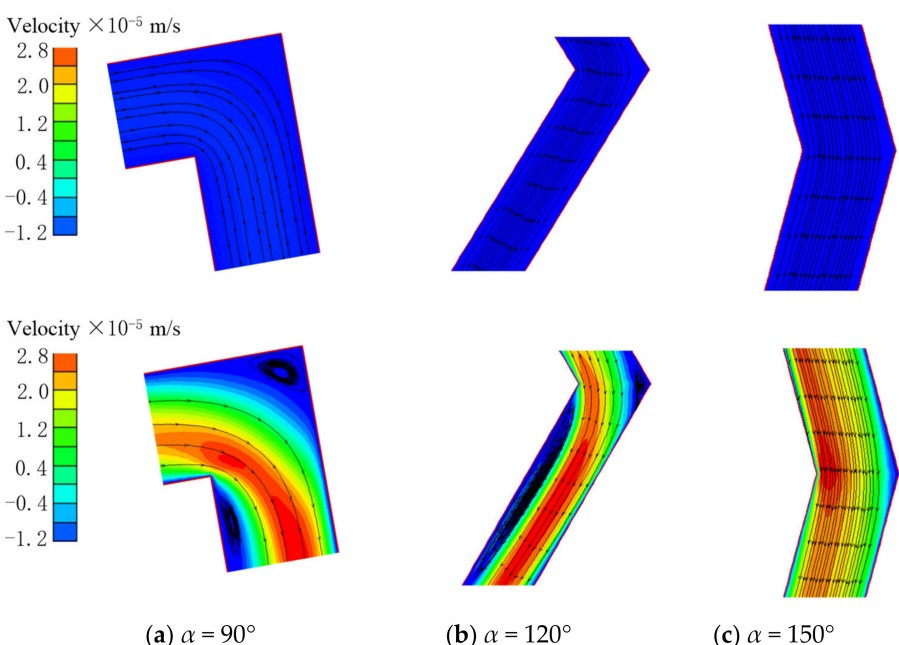

(**a**) $\alpha = 90°$      (**b**) $\alpha = 120°$      (**c**) $\alpha = 150°$

**Figure 9.** Flow velocity profiles for BFI at hydraulic gradient $J = 0.001$ (top row) and $J = 0.3$ (bottom row) [5].

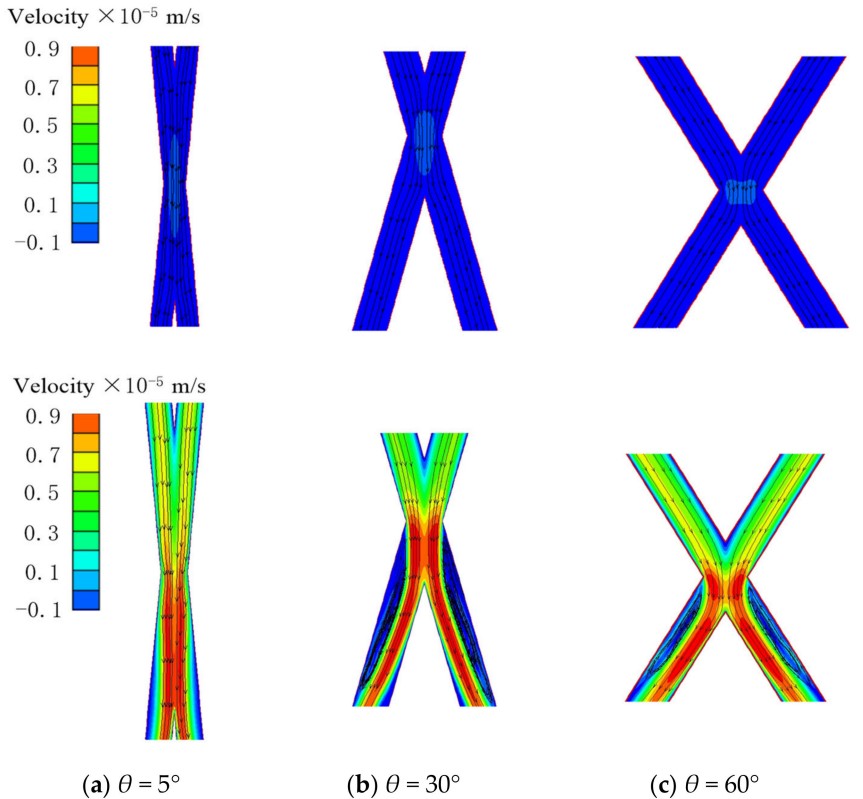

(**a**) $\theta = 5°$      (**b**) $\theta = 30°$      (**c**) $\theta = 60°$

**Figure 10.** Flow velocity profiles for CFI at hydraulic gradient $J = 0.01$ (top row) and $J = 0.3$ (bottom row) [5].

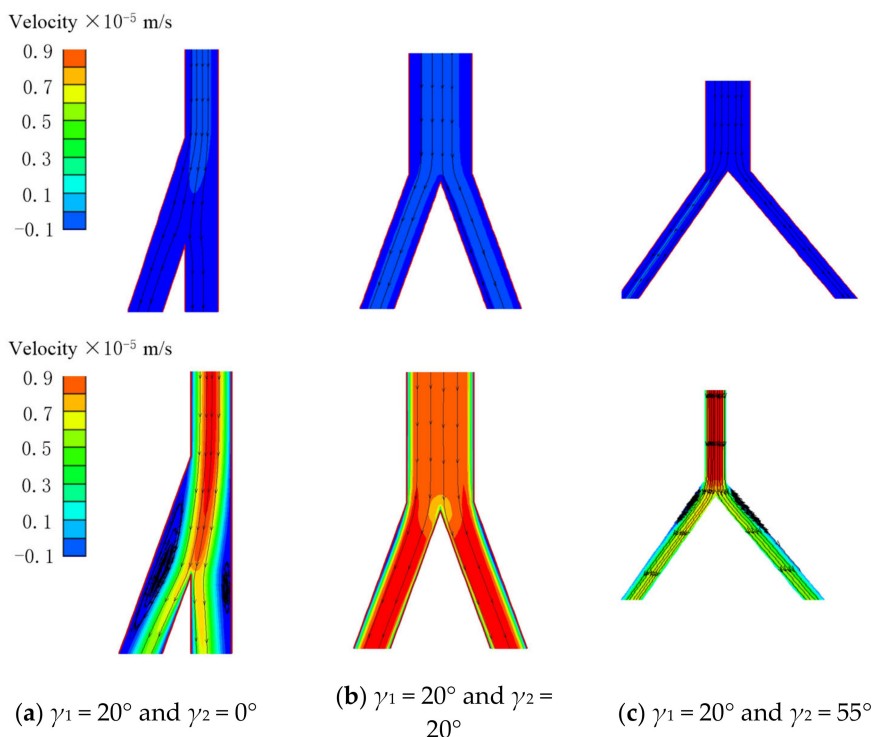

**Figure 11.** Flow velocity profiles for FFI at hydraulic gradient $J = 0.01$ (top row) and $J = 0.3$ (bottom row) [5].

Therefore, the formation of eddies, and hence the non-linearity of the flow, is related to the fracture intersecting angle, which demonstrates that the intersecting angles $\alpha$, $\theta$, and $\gamma$ play a significant role in the non-linear behaviour of flow in fracture intersections. This non-linear behaviour can be described by the Forchheimer equation (see Equation (22) in Section 3.4). The linear term ($AQ$) in the Forchheimer equation can be predicted by the Cubic law (see the previous section). However, the non-linear term ($BQ^2$) is not easy to determine due to the complex influences of the intersecting angle on the non-linearity of the flow. In our recent work [5], semi-empirical relationships between the non-linear coefficient $B$ of the Forchheimer equation and $\alpha$, $\theta$ or $\gamma$ of BFI, CFI, and FFI were proposed on the basis of extensive numerical simulations:

$$B = \frac{m_1 cos^2\left(\frac{\alpha}{2}\right) + m_2 cos^4\left(\frac{\alpha}{2}\right)}{Le^2w^2} \tag{17}$$

$$B = \frac{m_3 sin^2\left(\frac{\alpha}{2}\right) + m_4 sin^4\left(\frac{\alpha}{2}\right)}{Le^2w^2} \tag{18}$$

$$= \frac{m_5 sin^2\left(\frac{\alpha}{2}\right) + m_6 sin^4\left(\frac{\alpha}{2}\right)}{Le^2w^2} \tag{19}$$

where $w$ is the fracture width; $m_1$, $m_2$, $m_3$, $m_4$, $m_5$, and $m_6$ are regression coefficients (0.103, 4.138, 3.93, $7.37 \times 10^{-7}$, 38.4, and $7.38 \times 10^{-7}$, respectively, based on our simulation results); $e$ is the fracture hydraulic aperture as defined above and $L$ is total flow path length. Note these equations include the effects of both fracture intersecting angle and fracture aperture on the non-linear flow behaviour. More details of the analyses and descriptions of the flow behaviours at fracture intersections can be found in [5].

### 3.4. Non-Linear Fluid Flow Modelling in DFNs

Although there are abundant studies on modelling fluid flow in fractures and fracture networks, research on non-linear flow behaviours in DFNs is still limited. In the past

few years, we developed an advanced numerical procedure to address the non-linearity problem of flow through 3D DFNs [4].

As described in Section 3.2, the fluid flow within a fracture can be described by the Reynolds equation (RE) instead of the Navier-Stokes equation [102,103]. The governing equation can be written as:

$$\frac{\partial \rho g}{\partial x}\left(T_x \frac{\partial H}{\partial x}\right) + \frac{\partial \rho g}{\partial y}\left(T_y \frac{\partial H}{\partial y}\right) = 0 \tag{20}$$

where $H$ is the hydraulic head ($=P/\rho g$), $\rho$ is fluid density, $g$ is gravity acceleration, and $T_x$ and $T_y$ are fracture transmissivity in the $x$- and $y$-directions, respectively. Most previous works (e.g., [103,104]) assumed a linear relationship between flow rate and pressure gradient, following the cubic law, which defines transmissivity as:

$$T = e_h^3 / 12\eta \tag{21}$$

where $e_h$ is the hydraulic aperture, and $\eta$ is the fluid dynamic viscosity. However, for flow with high velocity (i.e., high Reynolds number), non-linear flow regimes would occur within fractures and fracture intersections, as demonstrated in the previous two sections. Zimmerman et al. [102] suggested that non-linear flow regimes in fractured media could be described by the Forchheimer equation:

$$-\nabla P = AQ + BQ^2 \tag{22}$$

where $A$ and $B$ are the linear and non-linear coefficients of the Forchheimer equation. By combining Equation (21) with Equation (22), the relationship between the transmissivity $T_x$ or $T_y$ and the flow rate $Q$ for isotropic case ($T_x = T_y$) is [102] can be expressed as:

$$T_x = T_y = \frac{-\eta Q}{\nabla P} = \frac{2\eta}{A + \sqrt{A^2 - 4B\rho g \nabla H}} \tag{23}$$

The Galerkin method is adopted to solve the non-linear problem of flow through 3D DFNs. Equation (20) can be rewritten using the equivalent weak integral form. The hydraulic head in each element satisfies the equation:

$$\int_\Omega \left(T_x \frac{\partial^2 H}{\partial x^2} + T_y \frac{\partial^2 H}{\partial y^2}\right) v(x,y) d\Omega = 0 \tag{24}$$

where $v(x, y)$ is the weight function.

Based on the Galerkin method, Equation (24) can be rewritten as:

$$\int_\Omega \left(T_x \frac{\partial^2 H}{\partial x^2} + T_y \frac{\partial^2 H}{\partial y^2}\right) d\Omega = \int_\Gamma T_n \frac{\partial H}{\partial n} v(x,y) d\Gamma \tag{25}$$

Using the Lagrange interpolation function: $H(x,y) = N_i(x,y)H_i$ and $v(x,y) = N_j(x,y)$ in element domain $\Omega$, a discretised finite element formulation of the integral can be obtained as:

$$\sum_{i=1}^{M}\sum_{j=1}^{M}\left[k_{ij}\right]\{H_i\} = \{Q\} \tag{26}$$

$$k_{ij} = \int_{\Omega_e}\left(T_x \frac{\partial N_i(x,y)}{\partial x}\frac{\partial N_j(x,y)}{\partial x} + T_y \frac{\partial N_i(x,y)}{\partial y}\frac{\partial N_j(x,y)}{\partial y}\right)dxdy \tag{27}$$

where $[k_{ij}]$ is the element transmissivity matrix, $N_i$ and $N_j$ are shape functions with three nodes in the integration, $M$ is the total number of nodes. Equation (26) includes the element transmissivity matrix calculated by Equation (27). Note that Equation (26) is a non-linear system of equations because $[k_{ij}]$ is related to $\{H_i\}$ when considering the Forchheimer law.

$T_x$ and $T_y$ can be obtained from Equation (23). The direct iteration method is adopted here to solve the equations. The preconditioned conjugate gradients (PCG) method was used to solve the large-scale sparse system of linear equations in each non-linear solution iteration step. More detailed descriptions of the method can be found in [4].

An example of flow modelling in a DFN under hydraulic gradients of 0.01 and 1 using the proposed method is demonstrated below. The mesh of the 3D DFN is shown in Figure 12. Figure 13a,b show the distributions of hydraulic heads within the DFN under hydraulic gradient = 0.01 and 1, respectively. Figure 14a,b show the corresponding contour map of the flow velocity. As can be seen from these profiles, a high hydraulic gradient does not affect the relative distributions of hydraulic heads, but it can result in high variability in velocity distribution. It also clearly demonstrates that the fracture network shows strongly preferential flow paths at high pressure gradients, where a small portion of the fractures within the network carry most of the flow passing through the system.

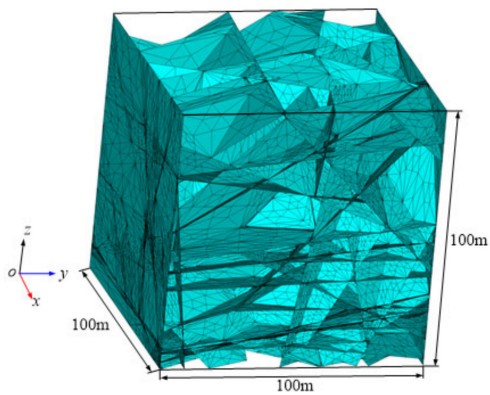

**Figure 12.** Mesh of 3D DFN.

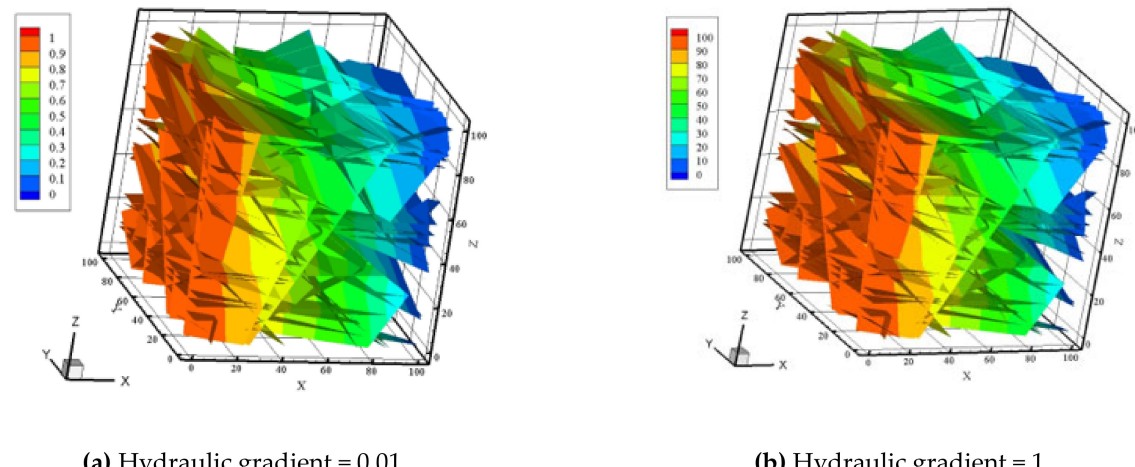

**(a)** Hydraulic gradient = 0.01  **(b)** Hydraulic gradient = 1

**Figure 13.** Pressure contour of 3D DFN calculated by the proposed method (unit: Pa) [4].

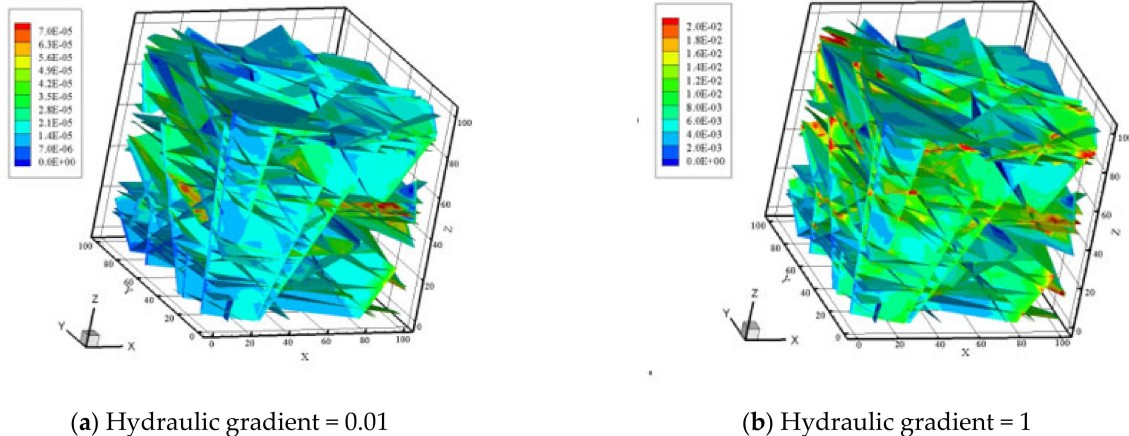

(**a**) Hydraulic gradient = 0.01                      (**b**) Hydraulic gradient = 1

**Figure 14.** Velocity contour of the 3D DFN calculated by the proposed method (unit: m/s) [4].

More cases of different DFN connectivity and fracture surface roughness were simulated. The results show that both the fracture network connectivity and fracture surface roughness play important roles in the non-linear flow characteristics in 3D DFNs. The transition from the linear flow regime to the non-linear flow regime occurs at a lower hydraulic gradient in the DFNs with smoother fracture surfaces and lower fracture connectivity. The critical hydraulic gradient ranges from 0.001 to 0.5.

*3.5. Experimental Studies of Non-Linear Fluid Flow through DFNs*

To understand the non-linear behaviours of fluid flow through fractures and fracture networks, we conducted numerous experimental tests on fractured rocks in different flow regimes over the past few years [4,19,96,105–107]. For the example given below, the rock material tested was sandstone collected from Hunan province in China. The sample tested is a cylindrical specimen with a diameter of 50 mm and a height of 100 mm. To create fractures in the specimen, the sample was subjected to an axial loading with a controlled axial displacement rate of 0.04 mm/min with no confining pressure using the RMT-301 rock and concrete mechanical testing system. The loading was stopped when fractures became visible on the specimen surface.

Before the flow experiment, X-ray micro-CT scanning of the rock specimen was carried out at a spatial resolution of 20 μm. An image processing method was developed to extract the details of the fracture network [19]. The details of the fracture geometries derived from micro-CT images are shown in Figure 15. The fracture porosity was found to be 1.85%, which was calculated as the ratio between the volume of the fracture void and the total volume of the sandstone specimen. There are 18 fractures with 28 intersections, and the dip angles of these fractures range from 20° to 80°. In addition, it can be seen that furcating fracture intersections are dominant in this case, followed by buckling fracture intersections (cf. Section 3.3). Each fracture is non-planar with rough wall geometries and clearly the spatial distribution of fracture apertures is non-homogeneous.

The relationship between the measured mean flow velocity $u_o$ in outlet and pressure gradient $\nabla p$ is shown in Figure 16. Clearly at low flow rates, the relationship is linear. When the flow velocity is greater than 0.02 m/s, equivalent to a Reynolds number of 0.5, the relationship becomes non-linear and the accuracy of Darcy's law in describing the flow process decreases. Using the Forchheimer equation to fit the experimental data, linear coefficient $a$ is 27157.0 kg/(m$^3$ s) and non-linear coefficient $b$ is 1098.5 kg/(m$^4$), respectively, in this case.

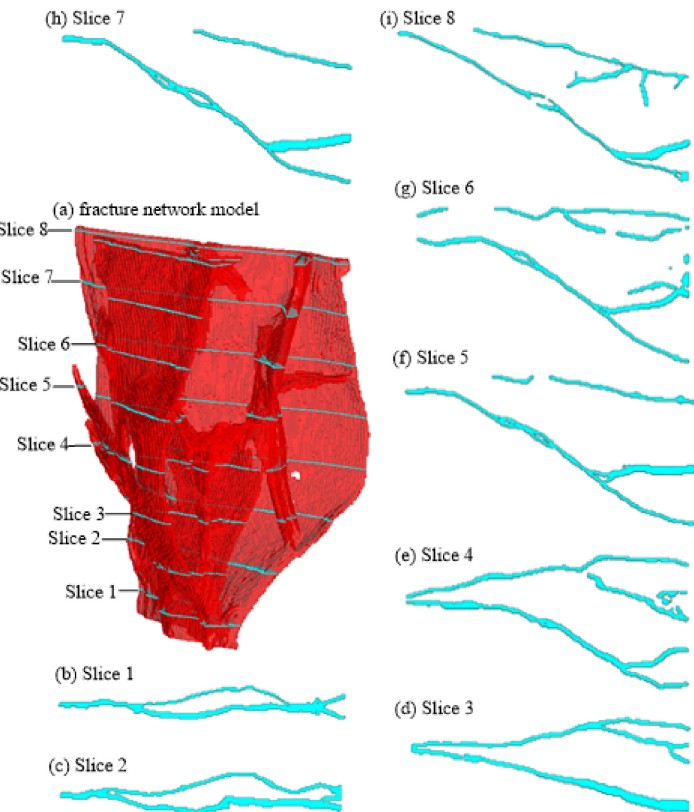

**Figure 15.** Fracture network specimen geometry and fracture intersections [19].

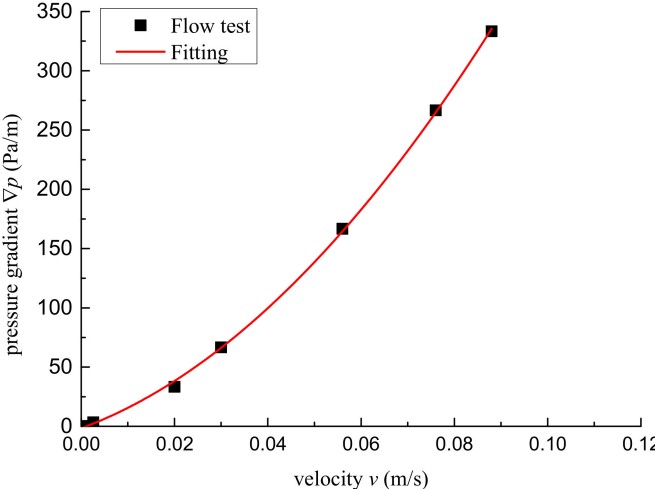

**Figure 16.** The relationship between the flow velocity and the pressure gradient for the sandstone specimen with 18 internal fractures and 28 fracture intersections [19].

Using the same geometries and flow conditions, ANSYS FLUENT was used to model the flow in the fractured rock specimen by solving the Naiver-Stokes equation. Figure 17 shows the streamlines of fluid flow through the fracture network at the cross-section of $y = 0.40$ when the hydraulic gradient is $J = 1 \times 10^{-5}$ and $1 \times 10^{-3}$. It can be seen that at $J = 1 \times 10^{-5}$, no eddy develops within the fractures and all the flow stream lines are parallel to the fracture walls. When $J$ increases to $1 \times 10^{-3}$, eddies emerge in fractures and fracture intersections. These eddies cause the reduction of the effective flow area within the fracture, and hence increase the non-linear behaviour of the flow. More discussions of the experimental results can be found in [19].

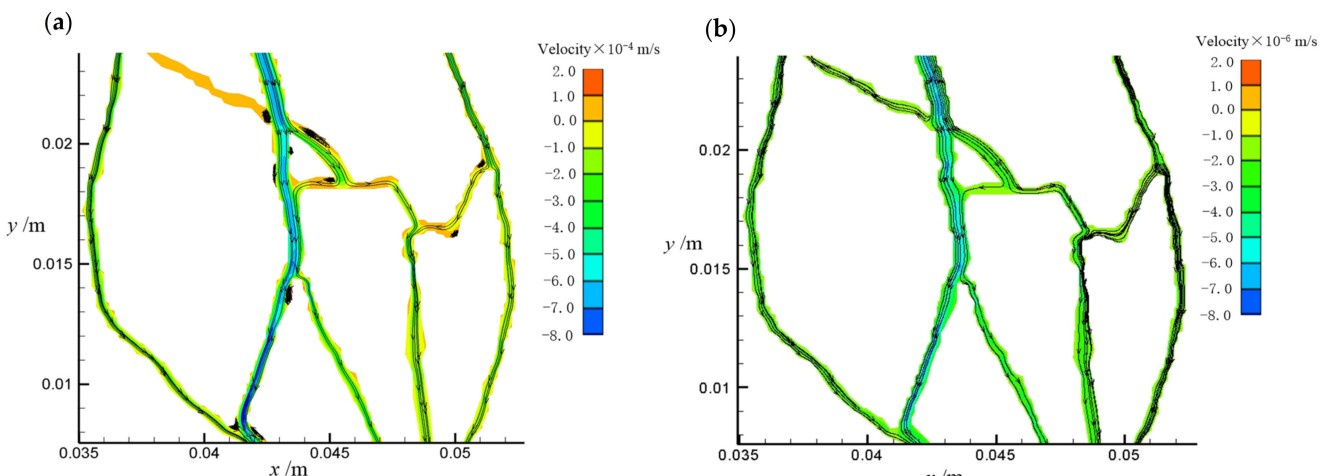

**Figure 17.** The flow velocity profile and streamlines on cross-section of y = 0.4 at different hydraulic gradients: (**a**) $J = 1.0 \times 10^{-5}$, (**b**) $J = 1.0 \times 10^{-3}$ [19].

## 4. Examples of Applications

Two application examples are briefly described in this section. The first one is the Habanero hot dry rock geothermal project and the second one is the in-situ recovery of copper minerals at the Kapunda mine site.

### 4.1. Habanero Geothermal Project

The Geodynamics Habanero reservoir in the Cooper Basin of South Australia is Australia's only hot dry rock heat resource exploited for geothermal energy. From 2003 to 2012, four wells (H1, H2, H3 and H4) were drilled in the area with depths ranging from 4221 m to 4421 m. The highest temperature recorded was 248.3 °C at 4391 m in H1, which suggests a likely bottom hole temperature in excess of 250 °C. Seven major hydraulic fracture stimulations were conducted during this period with a total volume of about 80,000 m$^3$ of stimulation fluid injected. The injectivity of the reservoir based on the fracture stimulation operations ranges from 1.0 to 16.0 l·MPa$^{-1}$·s$^{-1}$, depending on well locations. The stimulations helped create an enhanced geothermal system reservoir for the volume within which the wells were well connected. A short-term open-flow circulation test between H1 and H3 in 2008 achieved a flow rate of 18.5 kg·s$^{-1}$ with a temperature of ~212.5 °C at an injection pressure of 50.8 MPa. This was followed by a period of closed-loop flow testing between the two wells, which achieved a flow rate of 15.6 kg·s$^{-1}$ with a production temperature of ~212.5 °C and an outlet pressure of 44.8 MPa. A tracer test was also conducted, yielding a break-through time of four days and a peak return time of nine days. The mean residence time was 23.7 days and 78% of the tracer added was recovered. The open-flow testing between H1 and H4 achieved a maximum flow rate of 50 kg·s$^{-1}$ and the closed-loop circulation test in early 2013 achieved a maximum flow rate of 18.9 kg·s$^{-1}$ at the maximum production temperature of 215 °C. The 1 MWe electricity generation pilot plant was commissioned in April 2013 and the trial run of the plant lasted 160 days and concluded successfully in October 2013 [108].

During the fracture stimulation process, the surfaces of existing fractures can slip against each other due to the reduction in effective normal stress acting across the fractures; the misalignment of surface profiles results in dilation and hence increases the permeability of the fractures. New fractures can also be created during the stimulation process if the hydraulic pressure is high enough to overcome the minimum in-situ compressive stress. The Habanero basement rock is believed to be under an over-thrust stress regime and therefore shallow dipping pre-existing fractures will be critically stressed and can be stimulated with hydraulic pressure much lower than the vertical overburden stress. For these reasons, it is believed that the Habanero stimulations are mainly shear slip in nature

with some slips estimated to be in the order of centimetres based on the magnitude of the seismicity [108]. Slip between fracture surfaces or fracture initiation/propagation will produce micro-seismicity that can be detected by an array of geophones, from which the locations of events can be determined. It is then reasonable to assume that at least one fracture passes through any seismic event point, which provides a very useful means of conditioning fracture models to give more realistic models of the stimulated reservoir. Over 70,000 seismic event locations were determined for the seven major hydraulic stimulations discussed above. These event locations cover an area of approximately 4 km² with a vertical extent of around 500 m of the geothermal field.

Under the assumption that at least one fracture passes through each detected seismic point, it is possible, at least theoretically, to fit curved surfaces passing exactly through all seismic points, but such a solution soon becomes impossible to obtain in practice when the number of seismic points is significant and the fracture network is complex, as is the case for the Habanero reservoir. A compromise is to adopt the common approach in stochastic fracture modelling of using planar surfaces to represent fractures. As real fracture surfaces are tortuous, these planar surfaces can be considered as their first order approximate representations in three-dimensional space. A fracture model thus constructed can be viewed as a first-order representation of the reservoir that preserves all major features of the reservoir such as fracture connectivity and flow characteristics, which are important for the assessments of fluid flow in and heat extraction from the reservoir.

Over the past two decades, several techniques have been developed to fit a fracture model for the enhanced geothermal system reservoir conditioned on the seismic event points. One approach developed is the use of Markov chain Monte Carlo (MCMC) simulation to sample the extremely high-dimensional space to derive a set of fracture parameters; see details in Xu et al. [28] and Mardia et al. [109]. Two other approaches include the random sampling consensus (RANSAC) [69] and the clustering method [67,68]. A more elegant approach is the point and surface association consensus (PANSAC) approach [29], which effectively fits a stochastic fracture propagation (SFP) model. The SFP model attempts to follow the fracture propagation sequence during the reservoir stimulation process and has been demonstrated to produce a model much closer to reality than models generated by other approaches based on assessments of reservoir characteristics [29].

As an example, Figure 18 shows the seismic event cloud from the first major stimulation of the Habanero reservoir, which includes 23,232 point data. Figure 19 is the corresponding fracture model derived based on the MCMC approach.

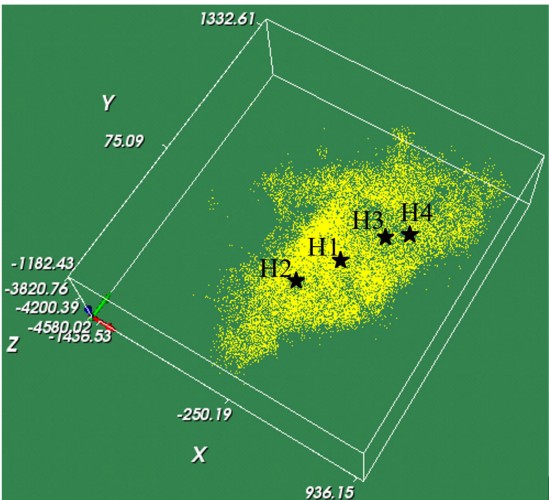

**Figure 18.** Absolute hypocentre locations of the seismic events.

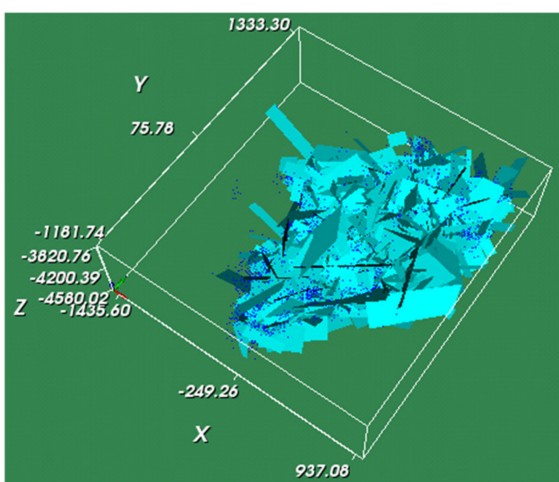

**Figure 19.** Habanero fracture model derived by MCMC optimization.

Conditional fracture models, such as those discussed above, are essential for realistic modelling of fluid flow and heat transfer in the reservoir. Early fluid flow and heat transfer models for the Habanero reservoir were largely based on an over-simplified single fracture model representing the major fault [110,111]. More realistic models address the complexity of the fracture network in the reservoir either by using the MINC concept [11] implemented in TOUGH2 [112] to model the fracture network [108] or by using a fracture model such as that discussed above [113]. The heat extraction assessment in this case is essentially a coupled hydro-thermal model, although the chemical and mechanical aspects may also have to be considered in cases where serious mineral scaling or changes in reservoir structure during production have to be considered (e.g., [82]). In Xu et al. [113], a simplified coupled HT model was developed for modelling geothermal heat extraction from an EGS reservoir. In order to model the full-scale reservoir operation of the geothermal field, an equivalent pipe network was implemented in their approach [17,18,113], which was demonstrated to produce results close to more accurate solutions obtained by the finite volume method but with only a fraction of the computation cost. Figure 20 shows the HT model and the predicted geothermal power generated from the H1-H3 doublet over a 20-year period using a flow rate of 35 $l \cdot s^{-1}$. Based on model assessments and flow circulation tests, it was concluded that a reasonable, conservative figure for the ratio of power to flow rate for the Habanero reservoir is 0.5 $MW \cdot l^{-1}$, which can be used to estimate the approximate geothermal power that could be generated from the reservoir [114].

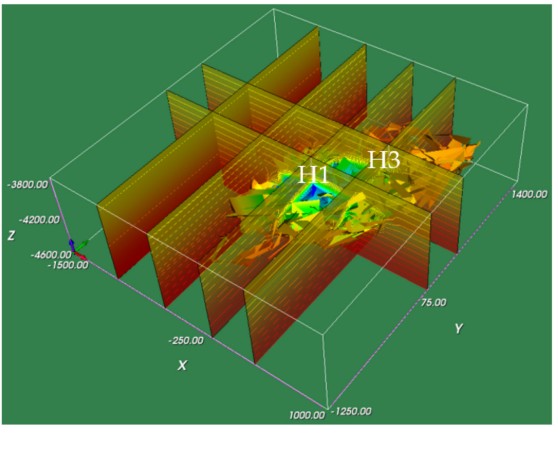

(**a**) H1-H3 doublet HT model

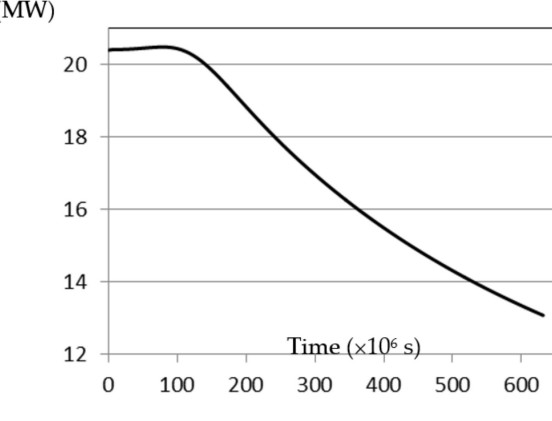

(**b**) Reservoir power generation (MW)

**Figure 20.** Reservoir HT model (**a**) and power generation from the reservoir (**b**).

### 4.2. In-Situ Recovery of Copper Minerals in Kapunda Mine

The Kapunda copper in-situ recovery (Cu ISR) project is the first non-uranium ISR project in Australia and is currently under a scoping study. The Kapunda area is highly fractured with copper mineralisation preferentially along fractures. It has an inferred resource of 119,000 tonnes of copper, which could be amenable to ISR [115]. One objective of the current scoping study is to understand the fracture development and the fracture networks in the Kapunda area, based on which, the flow pathways of the lixiviant through the fracture networks can be modelled and the recovery of copper can be predicted. For this purpose, three models have been developed, including a fracture model derived from the available fracture information, a hydraulic conductivity model based on the fracture model, and a coupled hydro-chemical model (HC model).

The fracture model was derived primarily on the basis of the fracture mapping of the open-pit mine that ceased operation in 1980 and the fracture density logging data collected from 59 preserved cores drilled within the area. The fracture density model (Figure 21) was first estimated from the drill core logging data using the Kriging method. The mine site was divided into four zones based on their slightly different fracture orientations derived from fracture mapping on the pit walls. Then a DFN for each block of the fracture density model was generated using the corresponding fracture orientation and size information for each zone. Both the DFN of each block and the large fault structures identified over the region (size exceeds common fracture size observed from the open pit) comprise the complete fracture model for the Kapunda area. The Kapunda old mine site also has underground workings from operations that ceased in 1877. These traverse the central area of the region and are highly conductive for groundwater flow.

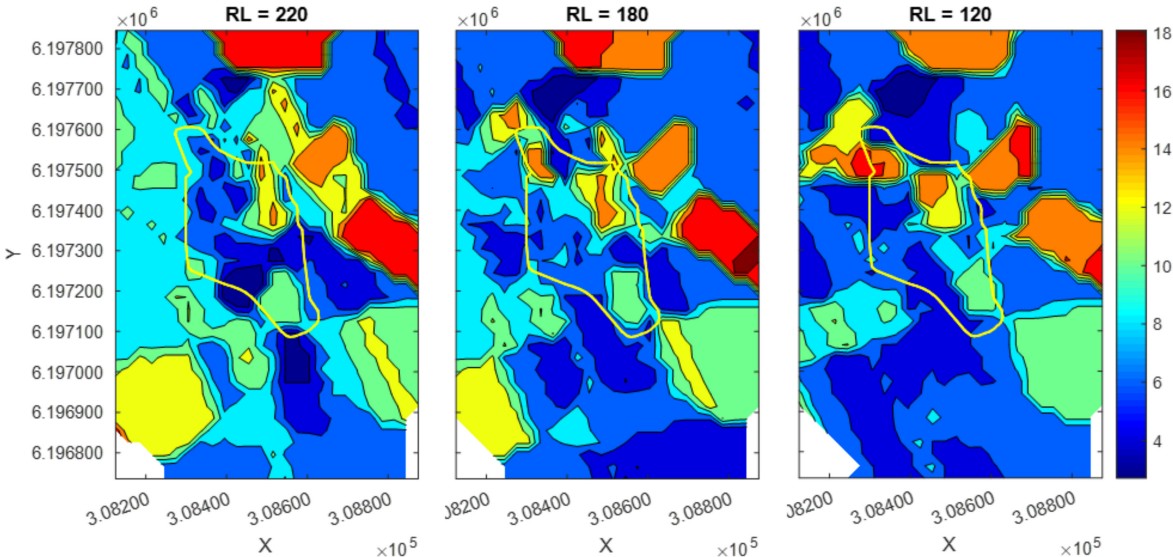

**Figure 21.** Horizontal cross-sections of the fracture density model at different relative levels (RL, m); coloured values represent the number of fractures per metre; the yellow string represents the footpath over the Kapunda old mine site (after Wang et al. [116]).

For the hydraulic conductivity model and the HC model, the underground workings and the identified large structures are modelled explicitly while the DFN of each block is treated as an ECPM (see Section 3.1) to avoid dealing with excessive numbers of fractures in the simulation. The explicitly modelled fracture geometries are treated as 2D surfaces instead of 3D solids due to their extensive two-dimensional, but narrow third dimension, shape, which, if modelled using a 3D approach, would cause meshing problems for the entire model region (see also Section 1). For the ECPM blocks, the permeability tensor can be determined by simulating fluid flow through the DFN of the blocks in COMSOL using the method described in Section 3.1. From the simulation results, a relationship can be

derived between the DFN permeability tensor and the fracture density for given fracture orientations and size distributions. Based on this relationship and the permeability tensor of the base-case DFN models, the hydraulic conductivity of all blocks can be estimated with correction factors applied considering fracture contacts and tortuosity. Figure 22 shows the estimated hydraulic conductivity together with the underground workings.

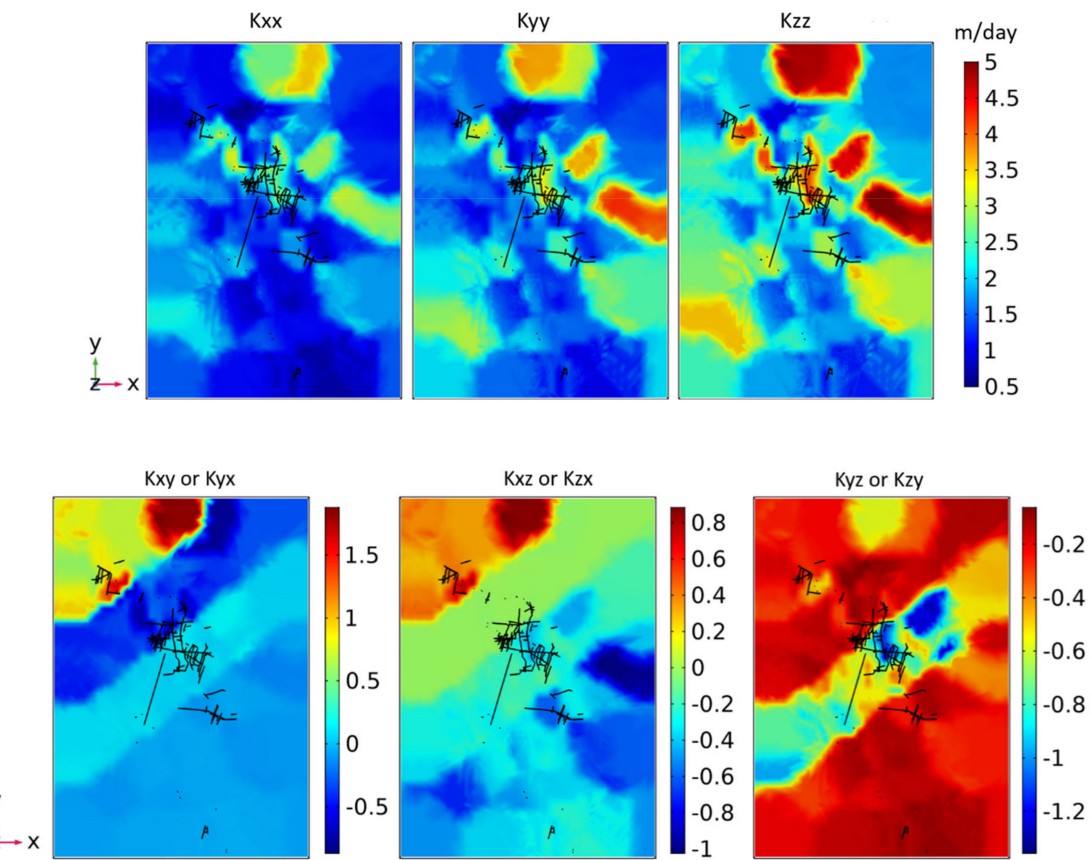

**Figure 22.** Horizontal cross-sections of the hydraulic conductivity model (in m/day) estimated for the Kapunda area (after Wang et al. [116]).

For the ISR operation, lixiviant is injected through wells into the mineralised areas and flows through fractures. In the process, lixiviant reacts with the contacted soluble copper minerals. The coper-bearing solution then flows to the production wells for extraction. The HC model derived in COMSOL simulates this process considering the reaction rate between Cu mineral and the lixiviant as well as the transport of the injected lixiviant and the pregnant leach solution (PLS) through both the ECPM domain and the explicitly represented large structures and underground workings. In this model, the volumetric reaction rate for Cu dissolution is given by the estimated surface area of copper minerals depending on the Cu grade and the surface reaction rate in first order kinetics with respect to acid concentration. The rate constant was estimated from the leaching test data using samples from the mine site. The injected lixiviant also reacts with the gangue minerals, which would affect Cu dissolution as it consumes acid. Therefore, the acid consumption by gangue minerals is considered in the model and the corresponding kinetics are also estimated from the leaching test data. Cu dissolution may also be affected by intro-aqueous geochemical reactions between various species in the solution. However, these processes are not modelled at the current stage of the scoping study. Depending on the metallurgical test results, if intro-aqueous geochemical reactions are found to have significant impacts on Cu dissolution, the modelling of these reactions can be incorporated in our model by coupling a geochemical modelling software package such as PHREEQC [117].

As an example scenario, Figure 23 shows a well-field design in which the short lines represent the screened/open section of the wells. Figure 24 shows the simulated copper concentration along the underground workings and at a horizontal cross-section after 3,500 hours of operation. The copper production curve for this example scenario is shown in Figure 25. The average copper grade in PLS from all extraction wells peaks at 2.6 g/L on day 83 and decreases below 0.05 g/L from day 854. If a cut-off grade of 0.05 g/L is used, the total copper production would be 3.95 Kt with a mine life of 854 days (2 years and 4 months). More details of this modelling work can be found in Wang et al. [116].

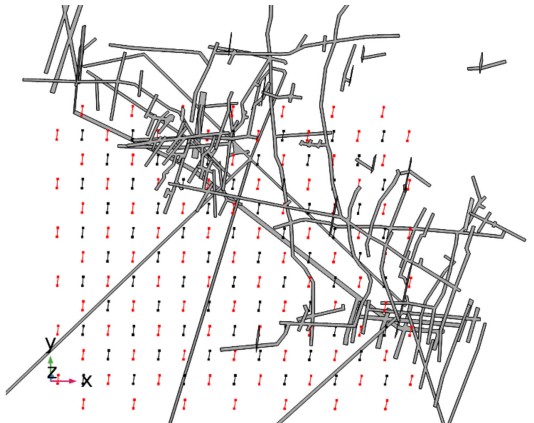

**Figure 23.** Well field design in an example scenario; red colour identifies the injection wells.

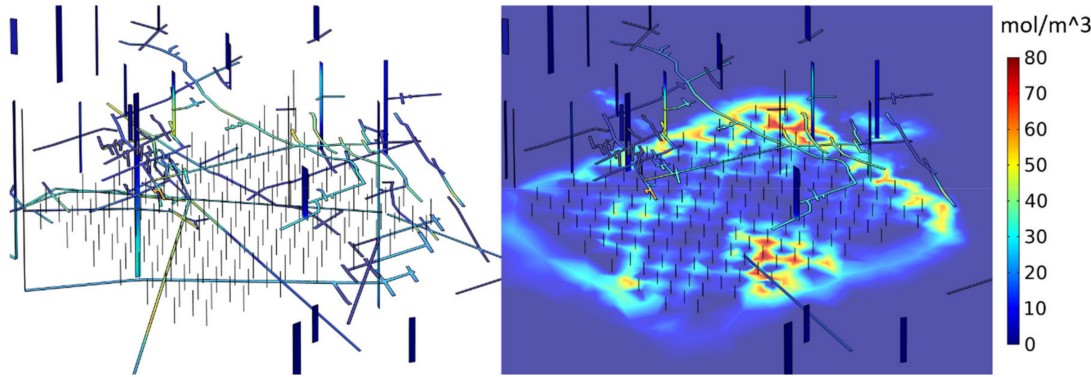

**Figure 24.** Simulated coper concentration in the underground workings and at a horizontal cross-section at the middle level of the open well sections after 3500 hours.

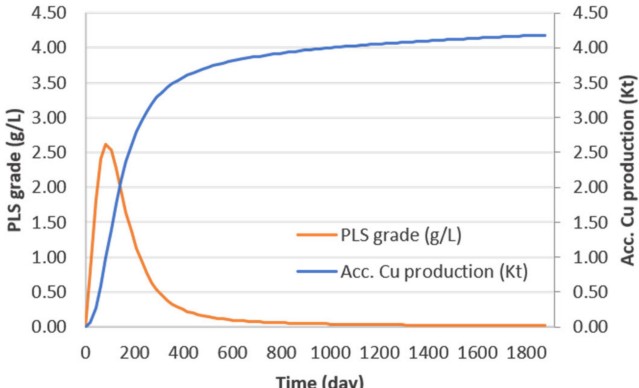

**Figure 25.** The predicted PLS grade and accumulated copper production for the example production scenario for the Kapunda copper ISR project.

## 5. Conclusions

Although the ideas remain largely unchanged, the modelling of rock fractures and fracture networks has undergone significant development over the past four decades. Various techniques that have been developed over that period are now available in commercial and research software packages that have the capacity to generate large-scale, sophisticated fracture systems more closely resembling reality in various applications. Over the course of these developments, the advance in the modelling of fluid flow through rock fractures and fracture networks has been even more significant, particularly in the past two decades, with many new methods and techniques proposed to address different problems. We have witnessed the progress of fracture network and flow models from over-simplified and mainly two-dimensional models in the early years to representative, large-scale, and fully three-dimensional solutions for rock engineering applications. In addition, methods and tools have been developed to incorporate different coupled conditions in the flow system, which more realistically reflect reality. Although there are still assumptions and simplifications involved in current solution frameworks, modern discrete fracture network modelling and its related application systems (e.g., flow modelling) have now matured in terms of accuracy, reliability, and capability to solve large-scale, complex engineering problems. Advances will continue to be made in the future development of DFN modelling with better incorporation of available conditioning data and indirectly sensed fracture information. We may also see an increasing use of machine learning in these applications.

This paper provides a brief review of fracture network modelling methods developed over the past few decades, including some recent developments from the international research group that has authored this paper. These developments include discrete fracture network modelling conditioned to seismic event point clouds in applications where the fracture network is created by fracture stimulation for, inter alia, enhanced geothermal systems, fractured reservoirs for the extractions of minerals, and unconventional gas or oil. The paper then covers some new advances made by this research group in modelling fluid flow through fractures, fracture intersections, and fracture networks. The focus of these new developments is mainly on the incorporation of challenging issues into flow modelling, including the considerations of the effects of fracture wall roughness, aperture variations, flow path tortuosity, fracture intersections, and inertia on the flow behaviours, particularly the non-linear characteristics of the flow through fractures and fracture networks. The methods developed have been demonstrated to be able to provide significantly improved flow predictions compared with conventional approaches, particularly in non-linear flow regimes. Finally, the paper gives a brief review of two practical applications for the methods developed. One is a hot dry rock geothermal heat extraction model, which involves a large-scale reservoir fracture model plus a hydro-thermal heat extraction model, and the other is the in-situ recovery of copper minerals from an old mine site, which involves a mine-scale fracture model and a hydro-chemical mineral extraction model that includes existing underground workings when modelling the flow.

The review provided in this paper is by no means comprehensive. Modelling of fractures and fracture networks and related applications has now broadened into many different areas, including mining, civil engineering, oil and gas extraction, hydrogeological engineering, geothermal systems, carbon sequestration, and in-situ recovery of minerals. These applications have their own particular characteristics and focuses that need to be specifically considered in modelling. The modelling of flow through, heat exchange in, and chemical reaction in the fractures and fracture network is just one of the important aspects of these systems. There is no doubt that this development will continue as there are still many problems related to fracture and fracture networks that remain to be solved.

**Author Contributions:** Authorships of this review paper: C.X. (Sections 1, 4.1 and 5), S.D. (Section 2), H.W. (Sections 3.1 and 4.2), Z.W. (Section 3.2), F.X. (Sections 3.3–3.5). All authors were involved in the planning, discussions and editing of this paper and all authors have read and agreed to the published version of the manuscript.

**Funding:** Australian Government's Co-operative Projects (CRC-P) Grants (CRCPFIVE000027), Australian Research Council Discovery Projects (DP110104766), Australian Research Council Industrial Transformation Training Centres Grants (IC190100017), Department of Primary Industries and Regions of South Australia (PIRSA), South Australian Centre for Geothermal Energy Research (SACGER), China Scholarship Council and University of Adelaide PhD Scholarships (201506430003), China Scholarship Council Visiting Scholarships (201706440044, 201806270149), National Natural Science Foundation of China (42002134), GuangDong Basic and Applied Basic Research Foundation (2020A1515111193).

**Conflicts of Interest:** The authors declare no conflict of interest. The funders had no role in the design of the study; in the collection, analyses, or interpretation of data; in the writing of the manuscript, or in the decision to publish the results.

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
