# Peer review of "Modelling of Coupled Hydro-Thermo-Chemical Fluid Flow through Rock Fracture Networks and Its Applications"

_geosciences, doi:10.3390/geosciences11040153_

Round 1

Reviewer 1 Report

This review addresses the recent development of fluid flow through rock fracture network, including modelling of fracture network, fluid flow in single fracture and fracture network, as well as engineering applications. This paper is well organized and well written. The review on numerical modelling is comprehensive, but that on experimental part can be further improved.

In Section 3.5, the authors should review more experimental works on the topic of non-linear fluid flow. Three papers reviewed here cannot give a comprehensive view. Some recent works can be included, such as, Yin et al. 2018. Rock Mech Rock Eng, 51, 3167-3177; Ji et al. 2020. Comp Geotech, 123, 103589; Dang et al. 2019. Comp Geotech, 114, 103152.

In the conclusions, the authors should address possible research directions of fracture network modelling based on the review work.

Author Response

This review addresses the recent development of fluid flow through rock fracture network, including modelling of fracture network, fluid flow in single fracture and fracture network, as well as engineering applications. This paper is well organized and well written. The review on numerical modelling is comprehensive, but that on experimental part can be further improved.

In Section 3.5, the authors should review more experimental works on the topic of non-linear fluid flow. Three papers reviewed here cannot give a comprehensive view. Some recent works can be included, such as, Yin et al. 2018. Rock Mech Rock Eng, 51, 3167-3177; Ji et al. 2020. Comp Geotech, 123, 103589; Dang et al. 2019. Comp Geotech, 114, 103152.

Response A1: Thanks for the suggestion. These papers have been reviewed and cited in Section 3.5.

In the conclusions, the authors should address possible research directions of fracture network modelling based on the review work.

Response A2: The following texts have been added to the Conclusions section to address this issue:

… Advances will continue to be made in the future development of DFN modelling with better incorporation of available conditioning data and indirectly sensed fracture information. We may also see an increasing use of machine learning in these applications.

Reviewer 2 Report

The manuscript is in general very well written and fit for publication after some revisions.

  • As the authors mainly discuss on the fracture network modelling, they should considerto change the title accordingly, e.g. “fracture network modelling for the coupled hydro-thermo-chemical numerical simulation applications” fits better. Authors may want to think about a new title.
  • The resolution of the figure is not of homogeneous quality, some figures can be replaced with high resolution versions. For example: Figure 6, Figure 13, Figure 14, Figure 15, Figure 25.
  • The same applies for the equations, whcih sometimes seem to be inserted as figues and sometimes using the equation tool in Word. Please check the font size and appearance of the equations and parameters to ensure a homogeneous high quality.
  • In section 3.1, where authors explain the ECPM, they represent a very simple example. It should be explained that such a permeability tensor for the whole system may not work nor reflect the reservoir behaviour in practice. Because, the fracture density and orientation are not homogeneous in the real fields and we need to grid the system to small subsections and obtain permeability tensors for these grid blocks. I think that COMSOL could not give such permeability tensors or it would be very time consuming. Therefore, I suggest that the authors introduce packages such as Fracman or DFNworks for equivalent permeability field calculation which provides more realistic reservoir permeability tensors.
  • In Figure 1, authors gathered a set of DFN modelling approaches. However, I saw some approaches in which researchers tried to consider some physical background of the fracturing process during fracture network modelling. If this approach is not considered within the categories of Figure 1, I recommend to mention them. See the following papers:

Masihi, M., King, P, R., “A correlated fracture network: modelling and percolation properties”, Water Resources Research, 43, W07439, 2007

Mahmoodpour, S., Masihi, M., “An improved simulated annealing algorithm in fracture network modelling”, Journal of Natural Gas Science and Engineering, 33, 538-550, 2016.

  • There is a problem in font formatting. When authors used “shown in”, the remaining part of the text goes to the next line. Please consider and change it accordingly. For example, lines 237- 243.
  • In Figure 9, the hydraulic gradient in the title is different than the figure. Also, I suggest to use the same hydraulic gradient for Figures 9- 11 or even a same color bar for better comparability.
  • Please use a same format for the heading of subsections. For example, compare 3.2 and 3.3.
  • It is better to have extra details in section 4.2. Which approach or software is used to obtain the hydraulic conductivity model (Figure 22)? Also, it is better to explain about the type of the chemical reactions or possible assumptions to model them with COMSOL. Because, most of the chemical reactions are kinetic controlled ones and we need to track the possible chain of reactions from reactants toward the products. COMSOL cannot handle this type of simulation. To handle this issue, researchers trying to couple geochemistry packages such as PHREEQC. Therefore, it is very important to discuss about the possible reactions and assumptions and the accuracy and draw backs of different implementation approaches.
  • Line 567: please cite a relevant paper about the SIMPLE algorithm.

Author Response

The manuscript is in general very well written and fit for publication after some revisions.

  • As the authors mainly discuss on the fracture network modelling, they should consider to change the title accordingly, e.g. “fracture network modelling for the coupled hydro-thermo-chemical numerical simulation applications” fits better. Authors may want to think about a new title.

Response B1: We appreciate your comment. However, we believe the title is appropriate and is a true reflection of the research reviewed in this paper. Fracture network modelling is only a small part of this review and the main part is actually dedicated to reviewing the modelling of coupled fluid flow.

  • The resolution of the figure is not of homogeneous quality, some figures can be replaced with high resolution versions. For example: Figure 6, Figure 13, Figure 14, Figure 15, Figure 25.

Response B2: Most of these figures have been replaced by higher resolution figures except for Figures 13 and 14, which are already in high resolution. The poor quality appearance was caused by conversion from WORD to PDF. We have also double-checked all other figures to ensure consistency in figure quality.

  • The same applies for the equations, whcih sometimes seem to be inserted as figues and sometimes using the equation tool in Word. Please check the font size and appearance of the equations and parameters to ensure a homogeneous high quality.

Response B3: Poor/inconsistent quality in equation typography was caused by conversion from a different version of WORD. We have now re-typed all equations using the same equation editor to ensure consistency in typographical quality.

  • In section 3.1, where authors explain the ECPM, they represent a very simple example. It should be explained that such a permeability tensor for the whole system may not work nor reflect the reservoir behaviour in practice. Because, the fracture density and orientation are not homogeneous in the real fields and we need to grid the system to small subsections and obtain permeability tensors for these grid blocks. I think that COMSOL could not give such permeability tensors or it would be very time consuming. Therefore, I suggest that the authors introduce packages such as Fracman or DFNworks for equivalent permeability field calculation which provides more realistic reservoir permeability tensors.

Response B4: We agreed with the issue raised about the ECPM approach. However, we do not claim ECPM will always work properly in all cases. In fact, as mentioned in the Introduction, ECPM is just one of several approaches commonly used to address the computational efficiency issue in modelling flow through fracture networks at the reservoir scale. In this section, we are reviewing the basic idea of the ECPM approach using COMSOL as the tool (which is also used in the second case study described in Section 4.2). The intention here was not to review or compare the implementation or accuracy of the ECPM approaches implemented in different software packages.

The following text has been added to the revised manuscript to address the issue raised by the reviewer:

The above example uses a small block to illustrate the process of estimating the permeability tensor of a DFN model using COMSOL. Such an estimation can also be performed using several other software packages, such as FRACMAN [111] and DFNWorks [112]. To use the ECPM approach at the reservoir scale, the region of interest can be sub-divided into blocks with their estimated equivalent permeability tensors. This approach is based on the assumption that sub-blocks satisfy the representative elementary volume requirement as described in Section 1. This could sometimes be challenging due to the inherent heterogeneity of fracture networks and therefore a certain degree of approximation will have to be used in practice.

  • In Figure 1, authors gathered a set of DFN modelling approaches. However, I saw some approaches in which researchers tried to consider some physical background of the fracturing process during fracture network modelling. If this approach is not considered within the categories of Figure 1, I recommend to mention them. See the following papers:

Masihi, M., King, P, R., “A correlated fracture network: modelling and percolation properties”, Water Resources Research, 43, W07439, 2007

Mahmoodpour, S., Masihi, M., “An improved simulated annealing algorithm in fracture network modelling”, Journal of Natural Gas Science and Engineering, 33, 538-550, 2016.

Response B5: Thanks for the suggestion. The two papers have been reviewed and cited in Section 2. The following text has been added to the revised manuscript:

Different correlation structures can also be considered in DFN modelling [109,110].

  • There is a problem in font formatting. When authors used “shown in”, the remaining part of the text goes to the next line. Please consider and change it accordingly. For example, lines 237- 243.

Response B6: This was a formatting error caused by the supplied WORD template. The problem has now been fixed.

  • In Figure 9, the hydraulic gradient in the title is different than the figure. Also, I suggest to use the same hydraulic gradient for Figures 9- 11 or even a same color bar for better comparability.

Response B7: Thanks for your careful review. The typo in Figure 9 has now been fixed. These figures have also been re-plotted using the same colour scale for easy comparison.

  • Please use a same format for the heading of subsections. For example, compare 3.2 and 3.3.

Response B8: Done. We have double-checked the manuscript to ensure consistency of style.

  • It is better to have extra details in section 4.2. Which approach or software is used to obtain the hydraulic conductivity model (Figure 22)? Also, it is better to explain about the type of the chemical reactions or possible assumptions to model them with COMSOL. Because, most of the chemical reactions are kinetic controlled ones and we need to track the possible chain of reactions from reactants toward the products. COMSOL cannot handle this type of simulation. To handle this issue, researchers trying to couple geochemistry packages such as PHREEQC. Therefore, it is very important to discuss about the possible reactions and assumptions and the accuracy and draw backs of different implementation approaches.

Response B10: The following text has been added in the revised manuscript to give more explanation of the issues raised in this comment:

In this model, the volumetric reaction rate for Cu dissolution is given by the estimated surface area of copper minerals depending on the Cu grade and the surface reaction rate in first order kinetics with respect to acid concentration. The rate constant was estimated from the leaching test data using samples from the mine site. The injected lixiviant also reacts with the gangue minerals, which would affect Cu dissolution as it consumes acid. Therefore, the acid consumption by gangue minerals is considered in the model and the corresponding kinetics are also estimated from the leaching test data. Cu dissolution may also be affected by intro-aqueous geochemical reactions between various species in the solution. However, these processes are not modelled at the current stage of the scoping study. Depending on the metallurgical test results, if intro-aqueous geochemical reactions are found to have significant impacts on Cu dissolution, the modelling of these reactions can be incorporated in our model by coupling a geochemical modelling software package such as PHREEQC [117].

  • Line 567: please cite a relevant paper about the SIMPLE algorithm.

Response B11: As suggested, the following reference is cited for the SIMPLE algorithm (Section 3.3):

Wang, H., Wang, H., Gao, F., Zhou, P. and Zhai, Z. (2018), Literature review on pressure–velocity decoupling algorithms applied to built-environment CFD simulation, Building and Environment, 143, pp. 671-678.